# Central American mountains inhibit eastern North Pacific seasonal tropical cyclone activity

Dan Fu [1,2✉], Ping Chang [1,2,3], Christina M. Patricola-DiRosario [4,5✉], R. Saravanan[1,3], Xue Liu[1,2] & Hylke E. Beck [6]

The eastern North Pacific (ENP) has the highest density of tropical cyclones (TCs) on earth, and yet the controls on TCs, from individual events to seasonal totals, remain poorly understood. One effect that has not been fully considered is the unique geography of the Central American mountains. Although observational studies suggest these mountains can readily fuel individual TCs through dynamical processes, here we show that these mountains indeed play the opposite role on the seasonal timescale, hindering seasonal ENP TC activity by up to 35%. We found that these mountains significantly interrupt the abundant moisture transport from the Caribbean Sea to the ENP, limiting deep convection over the open ocean area where TCs preferentially occur. This study advances our fundamental understanding of ENP TC genesis mechanisms across the weather-to-climate timescales, and also highlights the importance of topography representation in improving the ENP regional climate simulations, as well as TC seasonal predictions and future projections.

[1] International Laboratory for High-Resolution Earth System Prediction, Texas A&M University, College Station, TX, USA. [2] Department of Oceanography, Texas A&M University, College Station, TX, USA. [3] Department of Atmospheric Sciences, Texas A&M University, College Station, TX, USA. [4] Department of Geological and Atmospheric Sciences, Iowa State University, Ames, IA, USA. [5] Climate and Ecosystem Sciences Division, Lawrence Berkeley National Laboratory, Berkeley, CA, USA. [6] Department of Civil and Environmental Engineering, Princeton University, Princeton, NJ, USA. ✉email: fudan1991@tamu.edu; cmp28@iastate.edu

In the eastern North Pacific (ENP), about 17 named tropical cyclones (TCs) are generated annually on average, accounting for roughly 20% of the global total[1]. Nearly 75% of these ENP TCs form within a confined geographical area of 115–95°W and 8–16°N, making the ENP the most active basin in terms of TC activity density[2,3]. Such TC activity can have catastrophic socio-economic and ecological consequences on the southwestern United States, Mexico, the Hawaiian Islands, and maritime routes between these areas[4,5]. For example, the devastating flooding caused by Category 5 Hurricane Patricia (2015) and Hurricane Willa (2018) inflicted more than US $460 and $820 million in damages across Mexico and the United States[6,7]. As such, it is imperative to develop a fundamental understanding of the factors that control ENP TC activity in order to improve seasonal prediction and future projections of ENP TC activity and mitigate societal impacts.

ENP TCs are influenced by multiple factors on synoptic through seasonal timescales. On the synoptic scale, observations reveal that the majority of TCs in the ENP is directly triggered by or closely associated with tropical easterly waves (EWs), convectively coupled Kelvin waves, breakdown of the intertropical convergence zone (ITCZ), mesoscale convective systems, or the monsoon trough, through which lower-tropospheric cyclonic vorticity is maximized, making genesis more likely[8–13]. In addition, TC activity is substantially modulated on subseasonal-to-seasonal timescales by tropical modes of climate variability, such as the Madden-Julian Oscillation and El Niño-Southern Oscillation that can profoundly influence environmental vertical wind shear (VWS), deep convection, sea-surface temperature (SST), and mid-tropospheric moisture content over the ENP[14,15].

Besides the variety of atmospheric and oceanic factors that can modulate ENP TCs, a thorough understanding of the drivers of ENP TC genesis and activity should take into consideration the unique geography of the Central American mountains. These mountains extend along the Pacific coast from the Sierra Madre in North America and are on average ~1 km high. Importantly for TCs, the mountains are interrupted by three major gaps at the Isthmus of Tehuantepec, Gulf of Papagayo, and Panama Bight (Fig. 1a). Early theoretical and observational studies indicated that the favorable interactions of the Sierra Madre of Mexico, EWs, and the ITCZ could produce low-level cyclonic vorticity on the lee side, due to conservation of potential vorticity and additional vorticity advection from the ITCZ, which further promotes TC genesis[2,16–18]. Furthermore, the mountain gaps are associated with low-level jets at Tehuantepec and Papagayo, which can generate vorticity dipoles due to the sharp fan-like horizontal wind shear and additional counterclockwise flow curvature (Fig. 1b). However, there is no consensus on whether these cyclonic–anticyclonic vorticity dipoles would overall favor or inhibit TCs[17,19,20].

Given the limited observations and high computational cost of TC simulations, previous research on the influence of topography on ENP TCs has focused on synoptic dynamics, specifically, how the mountains generate the impinging TC precursors through dynamic processes. However, it is still unknown to date how this mountainous topography would influence the seasonal TC activity by altering the environmental TC favorability, especially from the role of broader climate thermodynamic states. This uncertainty becomes of key importance to the scientific community as many climate models still present significant biases of excessive rainfall in the vicinity of the ITCZ and consequent unfaithful seasonal ENP TC activity. Although the climate mean state biases may vary from model-to-model due to model physics parameterizations or dynamical core, recent studies[21–23] highlighted that these systematic model biases can also be partially attributed to the poor topographic representation, especially at mid-latitudes. Considering the current generation of climate models for climate prediction and projection typically have a relatively low resolution with heavily smoothed topography for numerical stability, one natural question arises: To what extent can the poor representation of topography lead to biases in the ENP simulation?

Here, we unravel the role of the Central American mountains in shaping ENP TC occurrence and spatial distribution on climate timescales, by conducting suites of TC-permitting climate simulations developed specifically for representing Northern Hemisphere TC activity and variability. The results show a significant interruption of the abundant moisture transport from the Caribbean Sea to the ENP by these mountains, which reduces deep convection over the open ocean area where TCs preferentially occur. As a result, seasonal ENP TC activity is hindered by up to 35% on seasonal timescales due to the presence of these mountains.

## Results

**High-resolution TC permitting climate simulations.** We performed numerical simulations using the atmosphere-only Weather Research and Forecasting (WRF[24]) model, which is developed by the National Center for Atmospheric Research (NCAR). Although we did not use an atmosphere–ocean coupled model in this study, the model is configured with a tropical channel domain and carefully tailored to represent the observed climatology and variability of ENP TCs and large-scale circulation with high fidelity[25]. Using the tropical channel domain enables us to determine the mountain's remote influences on various ocean basins.

Three sets of numerical simulations with a TC-permitting horizontal resolution of 27 km were performed, which compare simulations with no Central American mountains (hereafter referrred to as NMT) and no mountain gaps (hereafter referrred to as NGP) to reference control (hereafter referred to as CTL) simulation. For each set, simulations were conducted for 29 boreal summer seasons from 1990 to 2018 with a 6-member ensemble for each season, for a total of 174 members. The duration of each integration is ~7 months, and the model initial and lateral boundary conditions and SST forcing are prescribed using reanalysis. The model outputs are archived every 6 h, which enables the analysis across multiple spatial and temporal scales. In addition, the large number of simulations allows for the representation of atmospheric internal variability and the determination of statistical significance. CTL simulations use topography from the 2-min Gridded Global Relief Data (ETOPO2) of the US National Geophysical Data Center[26] that is interpolated to the model grid.

Supplementary Fig. 1 validates CTL with various observational data sets. Compared with observed high-resolution blended precipitation and outgoing longwave radiation (OLR), CTL reproduces a faithful mean location and strength of the ITCZ in the ENP (Supplementary Fig. 1a–g), where many climate models commonly overestimate the strength. Furthermore, CTL shows a broad agreement of precipitation in the Caribbean Sea, especially on the Atlantic slope of the Central American mountains, suggesting CTL reproduces reasonably well the easterly moisture transports. In contrast to the heavy precipitation on the windward side, both observations and CTL exhibit a weakened precipitation patch on the lee side of the mountains near the Costa Rica dome (11°N, 88°W), although CTL shows a positive rainfall bias. The near-surface circulation pattern over the Caribbean Sea and the ENP is consistent between satellite observations and CTL (Supplementary Fig. 1c–h), but CTL slightly overestimates the climatological strength of gap-wind jets

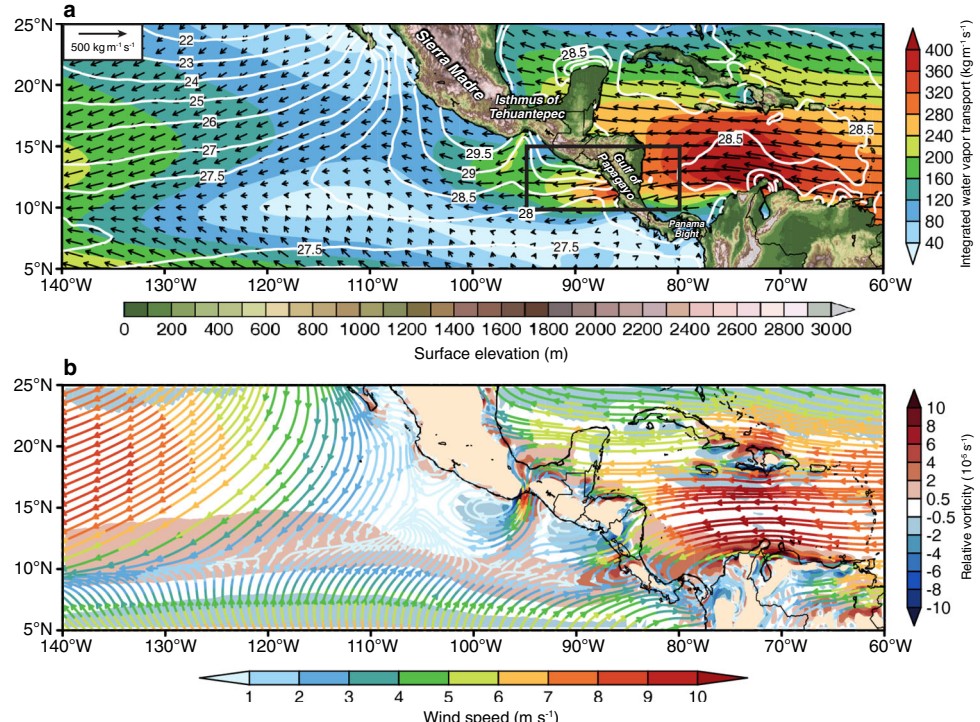

**Fig. 1 Thermodynamic and dynamic climate mean states from the observations. a** June–November 1990–2018 averaged NOAA Optimum Interpolation sea-surface temperature (NOAA OISST; °C) in white contours, ERA5 reanalysis data integrated water vapor transport magnitude (IVT; kg m$^{-1}$ s$^{-1}$) in color shadings (right color bar) and direction in vectors, overlaid with land surface elevation (m; bottom color bar). Black box (95°W–80°W, 10°N–15°N) illustrates the area of IVT intrabasin average in Fig. 6. **b** June–November 1990–2018 averaged ERA5 950 hPa relative vorticity (10$^{-5}$ s$^{-1}$; right color bar) in color shadings, wind speed (m s$^{-1}$; bottom color bar) and wind direction in streamlines. Note that yellow areas over land in (**b**) indicate the mountain area at 950 hPa.

and underestimates the cross-equatorial southerlies west of the Colombia Pacific coast. Related to these, CTL overestimates the latent heat flux near the Gulf of Tehuatepec, but overall shows remarkable agreement with the observation-based estimates (Supplementary Fig. 1d–i). This high fidelity of the simulated large-scale climate mean states helps ensure that seasonal ENP TC activity in the CTL is well simulated (Supplementary Fig. 1e–j).

In order to directly investigate the influence of the Central American mountains on ENP TCs on climate timescales, we performed two sets of simulations with modified topography. In NMT, we modified the Central American mountains between 95°W and 80°W to an averaged elevation of 50 m and land-use type of evergreen broadleaf, which are the predominant characteristics in the Yucatan Peninsula. In contrast, we filled the mountain gaps at the Tehuantepec and Papagayo area in NGP, while keeping the overall mountain orientation and land-use unchanged. We do not make changes at the Panama gap, as its influences on the TCs are minimal. The surface elevation for CTL, NMT, and NGP is shown in Fig. 2a–c.

To investigate uncertainty in the response of TCs to the Central American mountains due to model horizontal resolution and finer-scale topography, we performed additional higher-resolution downscaled CTL, NMT, and NGP simulations (referred to as CTL9km, NMT9km, and NGP9km), in which a 9 km resolution subdomain is one-way nested within the 27 km resolution tropical channel over the ENP and Gulf of Mexico, using a 30-member ensemble per each set (domain is shown in Supplementary Fig. 1a). The details of the model configuration, approaches to generate ensembles, and procedures for topography modification are described in "Methods."

**Seasonal ENP TC response to topography**. Boxplots of seasonal ENP TC activity from the 27 km tropical channel model simulations show a shift towards enhanced activity due to removing the Central American mountains and suppressed activity due to filling the gap-wind regions with terrain, relative to the real terrain case (Fig. 2d, e). NMT and NGP produce an ensemble mean seasonal number of TCs of 14.1 and 8.6, which are a statistically significant (5% level) enhancement (+27.6%) and reduction (−21.7%) from 11.0 TCs in CTL, respectively. Similarly, the accumulated days with ENP TC activity are also substantially increased (+33.0%) in NMT and decreased (−22.0%) in NGP relative to CTL, changing from 60.4 days in CTL to 80.1 in NMT and 47.0 in NGP, respectively.

The spatial distribution of the TC activity responses suggests that the Central American mountains exert complex influences on ENP TCs, which can extend far beyond the localized near-coastal area. In NMT, seasonal mean TC genesis and track density are reduced in the near-coastal ENP east of 95°W (Fig. 2g, j), whereas there is substantially enhanced TC activity in the rest of the ENP region. The maximum enhancement of genesis density occurs between 110°W and 100°W, whereas the greatest increase in track density occurs from 130°W to 105°W, both of which are near the area with peak TC activity (Fig. 2f, i). Combining the opposite sign of changes together, ENP TC activity in NMT is increased by about 35%. Similar to the NMT, NGP yields intrabasin variations in TC activity responses to the topography alteration. With the mountain gaps closed, there is a substantial decrease in TC activity over the majority of the ENP, except for an increase in TC activity near the Gulf of Tehuantepec, where the gap-wind jet exhibits clockwise (anticyclonic) curvature (Fig. 1h, k). Although the magnitude of the decreases in NGP is overall less than the

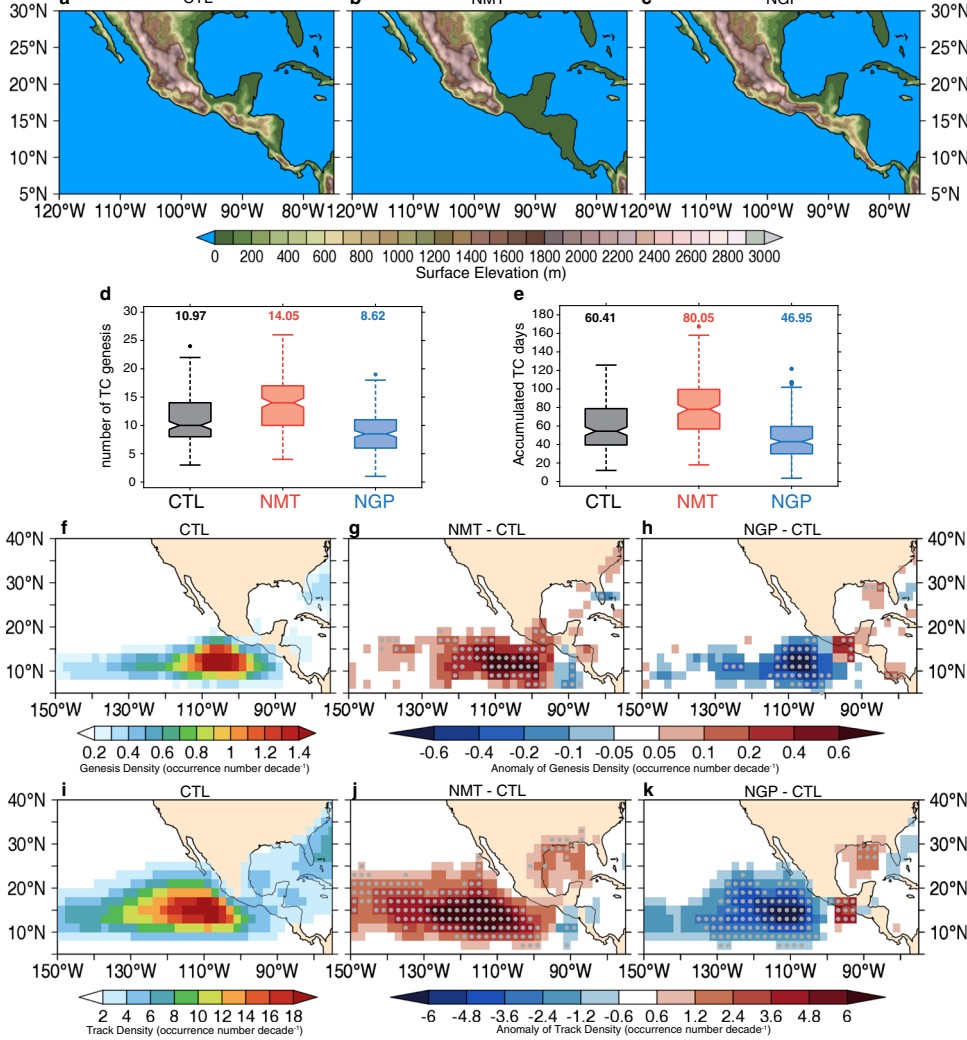

**Fig. 2 Seasonal TC activity variations due to topography.** Topography (m) used in **a** control (CTL), **b** no mountains (NMT), and **c** no mountain gaps (NGP) experiment. Boxplot of seasonal accumulated eastern North Pacific (ENP; 180°W to the North American coast, 0–30°N). **d** Number of TC genesis and **e** number of days with TC activity, showing the first quartile, median, and third quartile among CTL, NMT, and NGP simulations. The dashed vertical lines show the lowest and highest datum still within the 1.5 interquartile range (IQR), and the notch displays the confidence interval around the median. Outliers are denoted by dots, and the number on the top of each box indicates the mean value for that set of experiments. **f** CTL ensemble mean of seasonal TC genesis density (occurrence number decade$^{-1}$) and the differences using **g** NMT minus CTL and **h** NGP minus CTL. **i–k** Are similar, but for the TC track density (occurrence number decade$^{-1}$). Gray dots denote statistical confidence at the 95% level based on the two-sided $t$ test.

increases in NMT, the strongest changes still occur near 110°W, resulting in an overall reduction of seasonal TC activity by 22%.

We emphasize that the 9 km simulations produce nearly identical results in terms of percentage changes, with TC activity increased by about 36% in NMT9km and decreased by 25% in NGP9km relative to the seasonal average of 17.0 TCs and 95.2 TC days in CTL9km (Supplementary Fig. 2e–l). Although the spatial patterns of activity changes are less coherent owing to the smaller ensemble size, NMT9km and NGP9km again exhibit maximum changes concurrent with peak TC activity around 110°W, as well as the similar inverse sign of changes in the near-coastal ENP. These findings may seem unexpected when considering that the mountains can fuel TC genesis on the synoptic timescale; however, given the conceptual separation of weather and climate, and considering the near-identical results from various model horizontal resolutions with different capabilities in resolving TCs, there is strong evidence for the TC responses to the Central American mountains elucidated here, which are next explained by robust physical mechanisms.

**Topography-induced changes of dynamic and thermodynamic state.** To investigate the physical processes by which the Central American mountains influence ENP TCs, we disentangle the large-scale environmental factors that are linked to changes in TC genesis using a modified version of the genesis potential index (GPI[27,28]). The modified GPI well represents ENP TC genesis over the ITCZ-related strong convection regions (for details see "Methods"). The spatial correlation coefficient between seasonal TC genesis number and GPI improves from 0.73 using the original GPI to 0.83 in a modified version.

Consistent with the TC changes, NMT (NGP) produces a significant increase (decrease) in seasonal TC favorability over the majority of the ENP, except for opposite anomalies near the coastal area where the gap-wind jets are strong (Fig. 3b, c). By assessing the relative contributions of each term in the GPI calculation to the total (see "Methods"), we find that the more (less) favorable environmental conditions for ENP TC genesis in NMT (NGP) are primarily associated with changes in mid-tropospheric convective motion and relative humidity, whereas

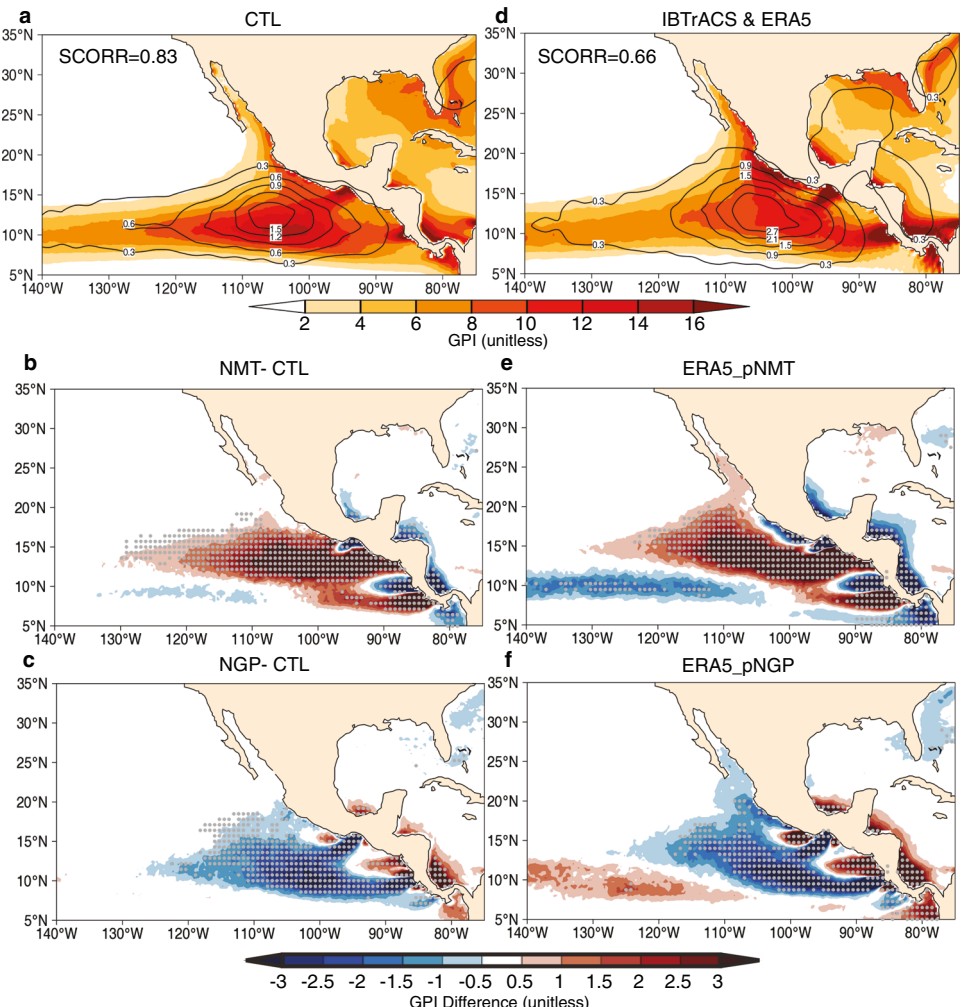

**Fig. 3 Topography modulates large-scale environmental favorability for seasonal TC genesis. a** Control (CTL) ensemble mean of genesis potential index (GPI; unitless; color shadings) and TC genesis density (occurrence number decade$^{-1}$; contour starting from 0.3 with the interval of 0.3). The differences of genesis potential index using **b** no mountains (NMT) minus CTL and **c** no mountain gaps (NGP) minus CTL. **d** Similar to (**a**), but for 1990–2018 mean genesis potential index from ERA5 reanalysis data (unitless; color shadings) and TC genesis density from the IBTrACS observation data (occurrence number decade$^{-1}$; contour starting from 0.3 with the interval of 0.6). The spatial correlation coefficients (SCORR) between TC genesis density and genesis potential index are listed on the top left of panels. **e**, **f** Similar to (**b**, **c**), but the GPI difference in "pesudo-no-mountains" (pNMT) and "pseudo-no-gaps" (pNGP) experiment (see details in "Methods"). Gray dots denote statistical confidence at the 95% level based on the two-sided t test.

the changes in VWS and lower-tropospheric vorticity dominate ENP nearshore area, where the regional circulation is strongly influenced by the mountain gap-wind variabilities (Fig. 4). These spatial patterns of anomalous GPI also appear in the "pseudo-no-mountains" (pNMT; see "Methods") and "pseudo-no-gaps" (pNGP; see "Methods") composites, using the ERA5 reanalysis[29]. The results suggest that atmospheric moist convection may play a key role in understanding why the Central American mountains significantly inhibit ENP TCs.

Seasonal mean precipitation, which is a proxy of atmospheric moist convection over the tropical ocean, exhibits anomalous patterns similar to the GPI responses (Fig. 5a–d and Supplementary Fig. 3b, c). A statistically significant increase (decrease) of precipitation is shown in a large portion of the ENP in NMT (NGP). Interestingly, precipitation over the Atlantic slope of the Central American mountains exhibits significant inverse changes compared to the ENP slope. Given that the Caribbean low-level jet transports abundant moisture from the Atlantic warm pool to the ENP (Fig. 1a) and triggers extensive orographic precipitation

(Supplementary Fig. 1a–f), a natural question to ask is: Can the Central American mountains exert a remote influence on ENP deep convection by blocking the upstream moisture transport?

We answer this question by estimating changes in vertical buoyancy[30] induced by temperature and specific humidity changes (see "Methods"). Figure 5b–e illustrates the changes in buoyancy in NMT and NGP along a transect across the Central America land bridge averaged between 8°N and 15°N. In NMT, buoyancy increases substantially over the original Central American mountains area (black shadings) from the lifted condensation level (LCL) all the way to the tropopause, accompanied by a westward extension within the lower-troposphere above the level of free convection (LFC) (Fig. 5b) due to the uninhibited moisture transport from the Caribbean Sea. The substantially enhanced buoyancy over the ENP increases the moist static instability; consequently, ascent above the LFC is significantly enhanced over the ENP (Fig. 5c and Supplementary Fig. 4j), which predominately contributes to the changes in TC activity. The changes are notably opposite in NGP: since the

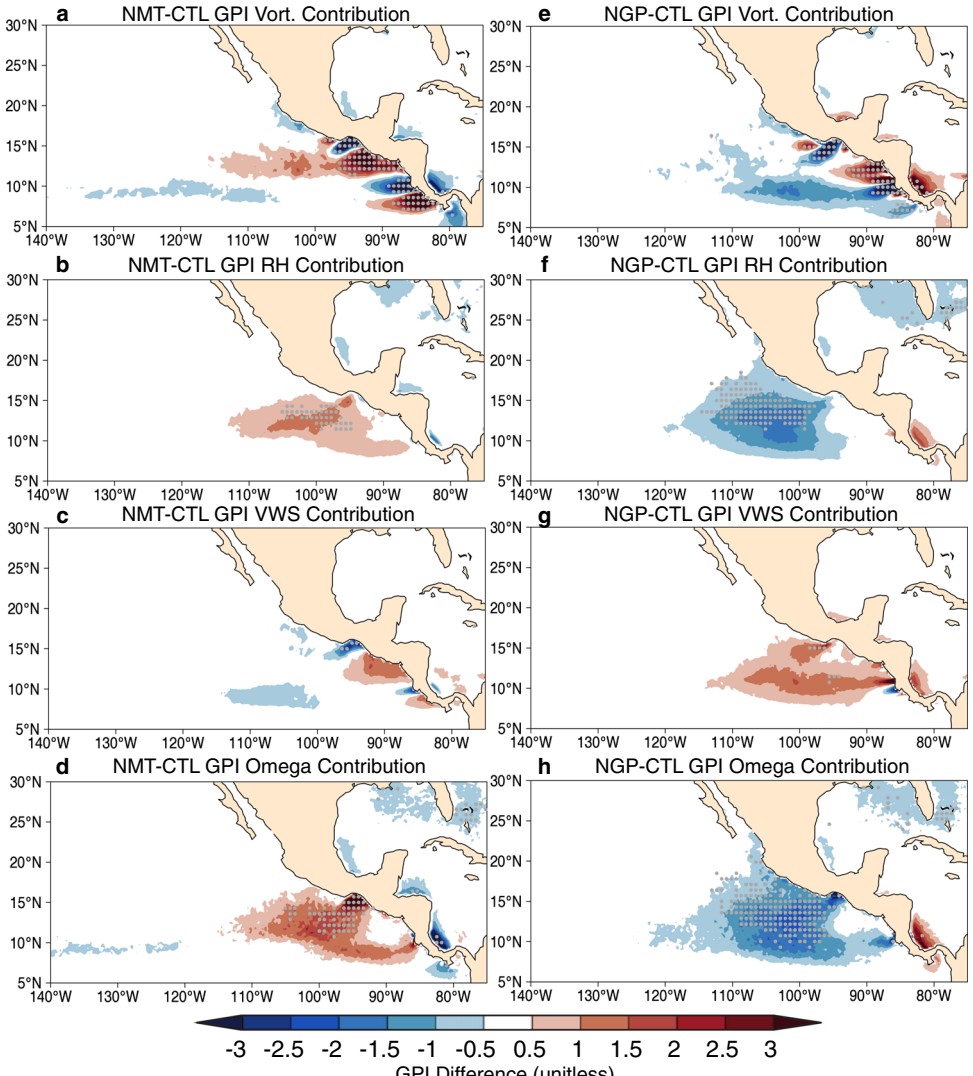

**Fig. 4 Estimation of relative contributions from each of modified genesis potential index terms responses to topographic change.** The June–November mean of the modified genesis potential index (GPI; unitless) calculated by setting the **a** 850 hPa vorticity, **b** 700 hPa relative humidity, **c** 200–850 hPa vertical wind shear, and **d** 500 hPa pressure level vertical velocity to values of no mountains (NMT) simulation, while setting other terms to values of control (CTL) simulation, minus the GPI directly derived from CTL simulation. **e–h** Similar, but for no mountain gaps (NGP) simulation. A positive anomaly indicates that environmental conditions are more favorable for TC genesis. We note that the relative contribution from the potential intensity (PI; m s$^{-1}$) term is very small and we do not show here.

mountain gaps are closed, the moisture transport convergence preferentially occurs over the windward slope due to the enhanced topographic lifting, resulting in more precipitation in this region, and thus drier conditions over the lee side make the air-column more stable and dampen convection over the ENP (Fig. 5e, f and Supplementary Fig. 4o), leading to suppression of the seasonal TC activity.

To better understand the role of this Atlantic-to-Pacific cross-basin moisture transport in altering convection over the ENP, we conduct a moisture budget analysis (refer to "Methods" for details). By decomposing the moisture flux convergence changes into the mean flow (monthly mean) and transient eddy (daily mean departures from the monthly mean) components, we found that the increased (decreased) precipitation in NMT (NGP) is predominantly controlled by the mean flow moisture flux convergence changes, while the transient eddy component plays a minor but negative role, indicating that high-frequency synoptic variability tends to transport the moisture from the equator to mid-latitude (Supplementary Figs. 5 and 6).

Meanwhile, we found that it is the dynamic contribution of anomalous monthly mean convergence of moisture flow that dominates the moisture budget in the ENP for both NMT and NGP (Supplementary Fig. 6). Supplementary Fig. 4 shows the mean circulation change for better illustration. It suggests that the further (reduced) extension of moisture transport by the seasonal mean winds from the Caribbean Sea into the ENP in NMT (NGP) accounts for the increased (decreased) mean flow moisture convergence, and thus increase (decrease) of the moisture content and moist convection in the ENP. The increased (decreased) moist convection further alter the large-scale relative vorticity by vortex stretching[3,31], and result in environmental conditions more (less) favorable for TC genesis (Supplementary Fig. 4).

The proposed mechanism is even consolidated through the evaluation of the anomalous ENP TC activity and area-averaged Atlantic-to-Pacific cross-basin integrated water vapor transport (IVT) over the Central American mountains (95°W–80°W and 10°N–15°N, shown in Fig. 1a). Seasonal mean boxplots reveal a

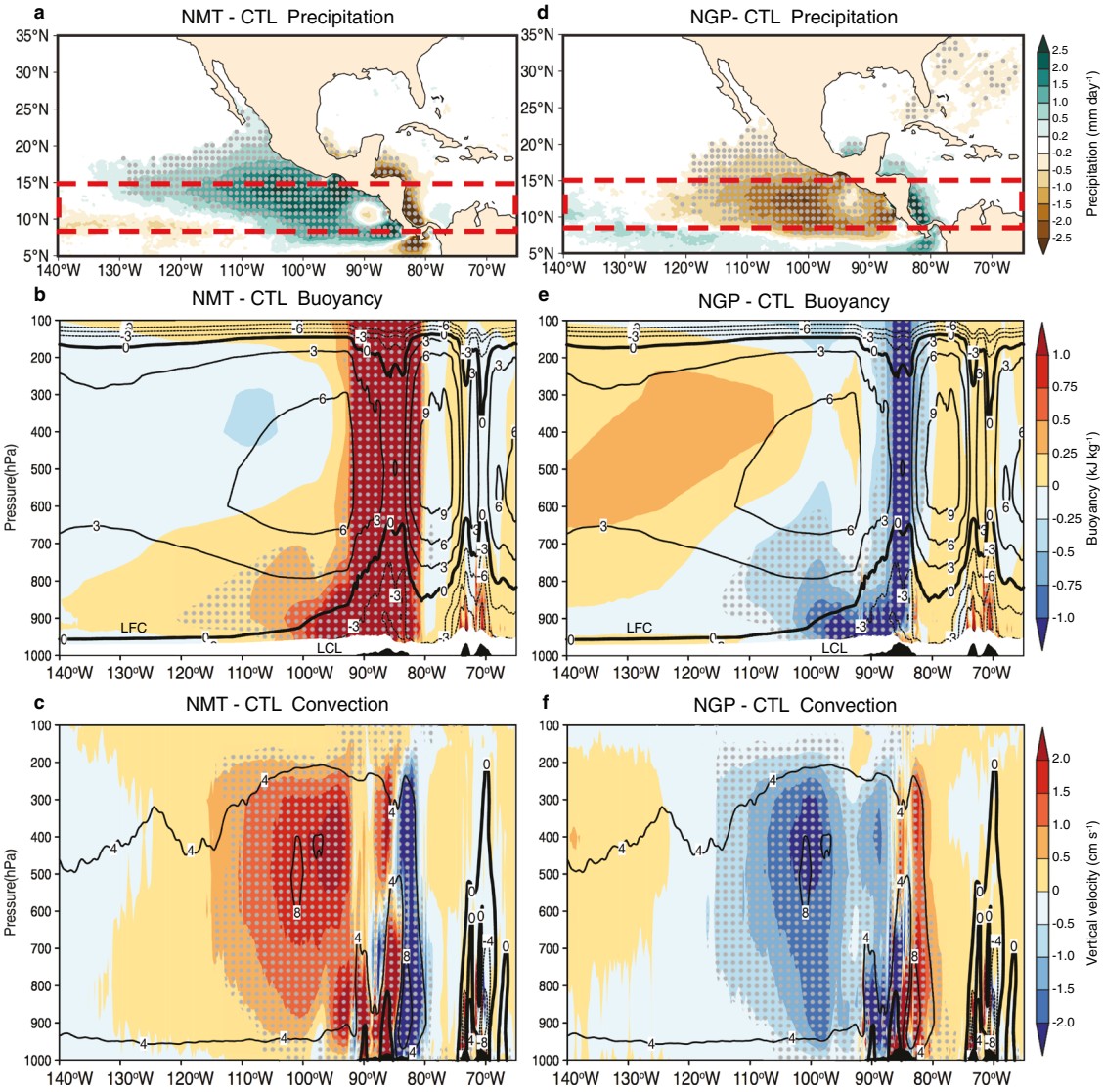

**Fig. 5 Topography-induced responses of thermodynamic climate states in the ENP. a** The precipitation difference (mm day$^{-1}$; color shading) using no mountains (NMT) minus control (CTL) ensemble mean. Vertical cross-section averaged from 8°N to 15°N [red dashed box in panel a] of **b** buoyancy (kJ kg$^{-1}$; positive upward; LFC the level of free convection, and LCL the lifted condensation level. Buoyancy values within LCL are masked out with white shadings as water vapor are not saturated. See "Methods" for details. **c** Vertical velocity (cm s$^{-1}$) difference (color shadings) using NMT minus CTL ensemble mean. CTL mean values are shown as contours. **d–f** Similar to (**a–c**), but for no mountain gaps (NGP) minus CTL. Gray dots denote statistical confidence at the 95% level based on the two-sided t test. Black shadings indicate the sketchings of averaged surface elevation.

robust linear regression between the increased (decreased) cross-basin IVT and the enhanced (suppressed) ENP TC activity in NMT (NGP) (Fig. 6a). In NMT, the significant changes in ENP TC activity occur in June–September, concurrent with the most salient anomalous cross-basin IVT (Fig. 6b, c). On the other hand, in NGP only June–August exhibits significant ENP TC activity and cross-basin IVT changes (Fig. 6b, c). As the area-averaged cross-basin IVT is calculated over the confined nearshore area that excludes the most significant TC activity changes located in the offshore ENP (see Fig. 2f–k), it indicates that cross-basin IVT causes the changes of TC activity, rather than the opposite causality. These results further support the role of the Central American mountains in suppressing ENP TCs, accomplished primarily by blocking the cross-basin moisture transport, which leads to dampened deep convection over the ENP.

Removal of the mountains' blocking effect on tropospheric moisture transport can induce anomalous diabatic heating over

the ENP, owing primarily to anomalous convection (Supplementary Fig. 7), and teleconnect with the other basins through equatorial wave propagation. However, this teleconnection only results in modest and insignificant changes in TC activity over the other TC-active regions (Supplementary Fig. 8), and its influences on TC intensity are minimal (Supplementary Fig. 9). Moreover, besides the influential modulation of the large-scale environmental TC favorability, the mountain-induced anomalous deep convection and diabatic heating can also affect the occurrence of the EWs[32] that originate locally in the ENP (Supplementary Fig. 10), which may also contribute to the anomalous ENP TC activity by altering the frequency of TC precursors.

## Discussion

By using suites of large-ensemble numerical experiments from a TC-permitting climate model tailored to simulate Northern Hemisphere TC activity and variability, we reveal the physical

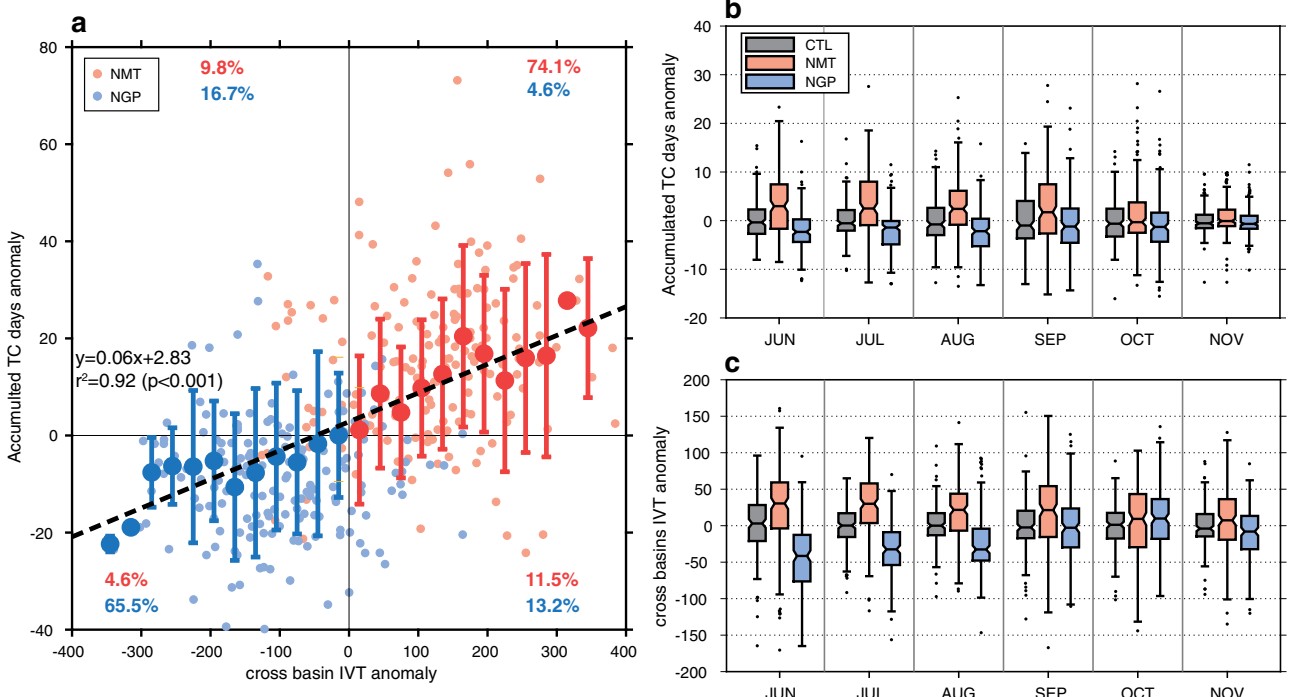

**Fig. 6 Sensitivity of the ENP TC activity to the Atlantic–Pacific interbasin moisture transport. a** The scatterplot (small transparent dots) of anomalous seasonal-averaged cross-basin integrated water vapor transport (IVT; kg m$^{-1}$ s$^{-1}$) and seasonal accumulated number of days with the eastern North Pacific (ENP) TC activity. Red dots denote no mountains (NMT) anomalies, while blue dots denote no mountain gaps (NGP) anomalies. The fractions within each quadrant are listed. The seasonal-averaged cross-basin IVT is area averaged over the box of 95°W–80°W, 10°N–15°N cross the Central America (shown in Fig. 1a). We note this box is carefully selected to exclude the area with significant TC activity changes in the offshore ENP. Binned scatterplots (large solid dots; intervals of 30 kg m$^{-1}$ s$^{-1}$) with the linear regression line are overlaid. Error bars are ±1 standard deviation of the binned-seasonal means. Boxplots of monthly variations of anomalous (**b**) accumulated number of the ENP TC days and (**c**) cross-basin integrated water vapor transport from control (CTL; black), NMT (red), and NGP (blue) simulations. The notch displays the confidence interval around the median and outliers are denoted by the dots. All anomalies were made by subtracting the corresponding season 6-member ensemble mean from CTL.

linkages through which local topographic features can influence extreme weather events on climate timescales. In particular, we advance a fundamental understanding of the role of Central American mountains in influencing the environmental conditions conducive to ENP TCs. We find that, although they can readily trigger TCs on synoptic timescales, the Central American mountains inhibit seasonal ENP TC activity by 35%, as the overall influence of the topography is to interrupt the abundant moisture transport from the Caribbean Sea, limiting deep convection over the open ocean area where TC density is highest. The responses in TC activity unraveled in this study are statistically significant and the underlying mechanisms are physically robust.

Our results are not contradictory with previous findings that focus on synoptic TC events that occurred at weather time scales. Instead, we broaden our fundamental understanding of how the Central American mountains influence TCs to climate timescales. Given the conceptual separation of weather and climate, one can understand our findings as follows: although the mountains can produce precursor atmospheric disturbances that act to trigger the development of individual TC events given sufficiently favorable environmental conditions, the climatological influence of the mountains is to make TC genesis less effective on seasonal timescales by dampening deep convection in the ENP.

Given that the salient TC activity responses to the Central American mountains are predominantly confined to June–September and over the broader ENP offshore area, the suggested strong air–sea interaction during boreal winter in the ENP, near the Costa Rica Dome and nearshore area[33], may have a limited influence. However, to fully explore the impact of

atmosphere–ocean coupling on the findings of this study, future coupled climate model simulations are required.

Another important implication of this study lies in the context of regional climate assessment. Previous study[34] and our results indicate that an accurate simulation and prediction of ENP TC activity on seasonal timescales rely on an accurate representation of the Atlantic-to-Pacific cross-basin moisture transport that is sensitive to resolution and local topography representation in prediction models. Several recent studies[35,36] further pointed out that the improved representation of topography is of particular importance in simulating regional circulation and hydroclimate. Combined with our results, it implies that the relatively low-resolution and heavily smoothed topography used in the current generation of climate models may be partially blamed for the common model biases in the ENP, especially the excessive precipitation and over-estimated convection in the vicinity of the ITCZ. Consequently, seasonal ENP TC activity and regional hydroclimate are not well represented in these climate models. These deficiencies raise concerns regarding the accuracy of such models in simulating changes in cross-basin moisture transport in response to climate change, which can potentially lead to uncertain projections of future ENP TC activity and regional circulations.

## Methods

**High-resolution climate model simulations.** The simulations in this study were run with the WRF regional climate model version 3.5.1, which is developed by the NCAR. The model was configured with 27 km horizontal resolution on a tropical channel domain covering 30°S to 50°N and carefully tailored for representing Northern Hemisphere TC activity and variability[25].

To provide a comprehensive assessment of statistical significance and for the purpose of increasing the ensemble spread, simulations were conducted for 29 boreal summer seasons from 1990 to 2018 with a 6-member ensemble for each season. Each simulation covers the main Northern Hemisphere hurricane seasons from June through November, initialized at different dates from 25 to 30 April, and ends on 1 December. The model output in April and May was discarded as spin-up. As such, in total, we produced a large-ensemble size of 174 members for 27 km resolution experiments. Initial and boundary conditions were taken from the 6-hourly National Centers for Environmental Prediction (NCEP) Climate Forecast System Reanalysis[37] for seasons before 2011, and from the NCEP Climate Forecast System version 2[38] afterward.

Due to the limited computational resources, the 9 km horizontal resolution one-way nested simulations (178°W–65°W, 2°S–36°N) were conducted with only a 30-member ensemble, and were performed in "climatology" mode[39]. The SSTs were prescribed with the 6-hourly climatology (1982–2018), while the lateral boundary conditions were derived from the 1996 season, which was characterized by a neutral phase of the Atlantic Multidecadal Oscillation. The initial conditions were from different dates from 1 to 30 April, and output from June to November was analyzed. The 9 km resolution simulations applied the same physical parameterizations as the 27 km resolution simulations; both are the same as in ref. [25]. We did not employ the topography-induced gravity wave drag parameterization in all simulations so that the subgrid-scale topographic asymmetry and convexity are not considered in this paper. We also did not apply any data assimilation techniques; all of the experiments were completed without any external forcing.

The criteria to identify and track TCs were obtained from a previous study[25]. The 850 hPa relative vorticity threshold for the 9 km resolution simulations was adjusted to $4.8 \times 10^{-4}\,\mathrm{s}^{-1}$, and kept as $1.6 \times 10^{-4}\,\mathrm{s}^{-1}$ for the 27 km resolution simulations. TCs were tracked using open-source TempestExtremes[40].

The 6-hourly detected TC center locations are counted within each $2° \times 2°$ grid box during each simulation season. The total count on the seasonal basis for each box is then defined as the TC track density. To reduce the noise, TC track density is smoothed by the 9-point moving average. The first detected location with at least tropical storm intensity ($\geq 17.5\,\mathrm{m\,s}^{-1}$) is defined as the TC genesis location, and the TC genesis density is defined similar to the TC track density.

**Experiment designs**. CTL)simulations for both the 27 and 9 km resolution simulations used real topographic height from the 2-min Gridded Global Relief Data (ETOPO2) of the US National Geophysical Data Center that is interpolated to the target model grid. To assess the model's ability to faithfully reproduce the key aspects of regional circulations in the ENP, the CTL simulations were validated with a variety of observations, including 6-hourly TC best track data from International Best Track Archive for Climate Stewardshi[41] version 4, surface wind analyses from Cross-Calibrated Multi-Platform version 2[42] surface wind vector analyses, precipitation from Multi-Source Weighted Ensemble Precipitation version 2.2 (MSWEPv2.2[43]), ocean surface latent heat flux from Japanese Ocean Flux Data Sets with Use of Remote-Sensing Observations version 3 (J-OFURO3[44]), and OLR from National Oceanic and Atmospheric Administration (NOAA-OLR[45]).

In the idealized NMT simulations, the interpolated ETOPO2 data are modified such that elevations are around 50 m over land areas between 95°W and 80°W. Meanwhile, we change the original various land-use types of the mountainous area to constant evergreen broadleaf; as such, the Central American mountains are numerically diminished. In addition to the NMT, we numerically fill the gaps near Tehuantepec and Papagayo to perform the NGP simulations. Due to the west–east orientation of the Tehuantepec gap, the interpolated ETOPO2 surface elevations in a rectangular box of 97–91°W and 16.5–17.5°N are modified. At each grid point within the box, we first defined the latitude-dependent critical values as box zonal mean plus one standard deviation of the zonal values, then the elevations lower than the latitude-dependent critical values were multiplied by a random factor in the range between 1 and 1.05 repeatedly until the modified elevations were larger than the critical values. To smooth the modified elevations, we also applied a two-dimension Gaussian filter with a size of 5 (15) zonal and 3 (9) meridional grid cells for 27 (9) km simulations and a standard deviation of 0.5. As a result, the mountain gap at Tehuantepec was entirely closed, whereas the non-gap regions were largely unchanged (Fig. 1c and Supplementary Fig. 2d). Due to the northwest-to-southeast orientation of the Papagayo gap, a similar approach was conducted over the Papagayo region, except a parallelogram (90.5°W, 15.7°N; 88.1°W, 15.7°N; 80.8°W, 8.6°N; 83.2°W, 8.6°N), rather than a rectangular area of surface elevations were modified.

**Modified GPI**. To better understand the physical processes by which the Central American mountains influence ENP TCs, we disentangle the large-scale environmental factors that are linked to changes in TC genesis using a modified version of the GPI, following a previous study:[28]

$$\mathrm{GPI} = |10^5 \eta|^{3/2} \left(\frac{\mathrm{RH}}{50}\right)^3 \left(\frac{V_{\mathrm{pot}}}{70}\right)^3 (1 + 0.1 V_{\mathrm{shear}})^{-2} \left(\frac{-\omega + 0.1}{0.1}\right) \quad (1)$$

where $\eta$ is the absolute vorticity ($\mathrm{s}^{-1}$) at 850 hPa, RH is the relative humidity (%) at 700 hPa, $V_{\mathrm{pot}}$ is the maximum potential intensity[46,47] ($\mathrm{m\,s}^{-1}$), $V_{\mathrm{shear}}$ is the magnitude of VWS ($\mathrm{m\,s}^{-1}$) between 850 and 200 hPa, and $\omega$ is the pressure coordinate

vertical wind velocity ($\mathrm{Pa\,s}^{-1}$) at 500 hPa. The modified GPI shown in Eq. (1) has a vertical wind velocity term that enables correct reproducibility of TC genesis over regions with strong convection, such as in the ENP.

The GPI was computed from monthly averaged model outputs. Compared to the original version of GPI, the spatial correlation coefficient between the CTL simulated seasonal TC genesis number and GPI improved from 0.73 to 0.83 in the modified version. To estimate the primary factors that support GPI responses in the NMT (NGP) simulations, the GPI was also calculated by setting one term to the value from the NMT (NGP), while setting the others as values from the CTL simulation, as in previous studies[39,48].

In addition to the direct model simulations of GPI responses to the Central American mountains, to further consolidate the proposed physical mechanism, we also performed the adjusted GPI analysis using state-of-the-art ERA5 reanalysis[29] with the pseudo-no-mountains (ERA5_pNMT) and pseudo-no-gaps (ERA5_pNGP) approach, which are essentially analogous to the pseudo-global warming method[49] that aimed to study climate change. Specifically, we used the three-dimensional monthly averaged temperature, specific humidity, relative humidity, the vertical and horizontal winds from the 174-member ensemble mean of the NMT and NGP to subtract the corresponding variables from the CTL ensemble mean, and derived the anomalous thermodynamical and dynamical climatology states. These monthly climatological anomalies were then linearly added onto the interannually varying monthly ERA5 data, to yield simple surrogate scenarios that the Central American mountains were diminished (ERA5_pNMT) or mountain gaps were filled (ERA5_pNGP). The newly constructed monthly data were then used to apply the aforementioned GPI analysis. Given that GPI is a complex metric determined by the products of multiple subset terms, and since the relative contributions from each term were also estimated nonlinearly, how the linearly added anomalous signals will influence the net GPI results is indeterminate.

**Buoyancy diagnostics**. The approach to estimate the buoyancy of a moist air parcel was adapted from a previous study[50]. Specifically, the buoyancy of a saturated convective air parcel, which is determined from the difference between its own temperature $T$ and environmental temperature $T_{\mathrm{env}}$, is proportional to the difference between the environmental saturation moist static energy and the ascending air parcel moist static energy:[30]

$$C_p(T - T_{\mathrm{env}}) = \frac{h - h^*_{\mathrm{env}}}{1 + \gamma} \quad (2)$$

where $h = C_p T + gz + Lq$ is the moist static energy of the ascending air parcel ($\mathrm{J\,kg}^{-1}$), and $h^*_{\mathrm{env}}$ is the environmental saturation moist state energy, $q$ is the specific humidity ($\mathrm{kg\,kg}^{-1}$), $g$ is the gravitational acceleration ($\mathrm{m\,s}^{-2}$), $C_p$ is the isobaric specific heat of dry air ($1004\,\mathrm{J\,K}^{-1}\,\mathrm{kg}^{-1}$), $L$ is the latent head of condensation ($2.5 \times 10^6\,\mathrm{J\,kg}^{-1}$), $\gamma = (L/C_p)(\partial q^*/\partial T)_p$, and $q^*(T,p)$ is the saturation specific humidity ($\mathrm{kg\,kg}^{-1}$) that is determined by August–Roche–Magnus formula[51]. Since the ascending air parcel is quasi-adiabatically lifted from near surface, and thus the moist static energy is roughly conserved during the ascending motion, one can approximate $h$ by the near-surface moist static energy, as such:

$$h \approx h_{10m} = C_p T_{10m} + gz_{10m} + Lq_{10m} \quad (3)$$

where $z_{10m} = 10$ m. Given that parameter $\gamma$ has an order of positive 1[30], the difference of $h_{10m} - h^*_{\mathrm{env}}$ is roughly equal to twice the buoyancy value. As the result, we estimate the three-dimensional buoyancy as $b = (h_{10m} - h^*_{\mathrm{env}})/2$ above the LCL, and the anomalous buoyancy in the NMT and the NGP relative to the CTL can then be determined. Note that the relationship between buoyancy and moist static energy within LCL does not hold as water vapors are unsaturated, thus the buoyancy differences within LCL are masked out with white shadings in Fig. 5.

**Moisture budget analysis**. To further understand whether dynamic or thermo-dynamic components of the moisture transport drive the changes of the moisture in response to the Central American mountains, we conducted the moisture budget analysis following the previous studies[52–55]. The seasonal mean atmospheric moisture budget equation can be written as:

$$\rho_w g(P - E) = -\int_0^{P_s} \nabla \cdot (\mathbf{u}q)\mathrm{d}p = -\int_0^{P_s} \nabla \cdot (\bar{\mathbf{u}}\bar{q})\mathrm{d}p - \int_0^{P_s} \nabla \cdot (\overline{\mathbf{u'}q'})\mathrm{d}p - q_s \cdot \mathbf{u}_s \nabla p_s \quad (4)$$

where $\rho_w$ is the water density ($\mathrm{kg\,m}^{-3}$), $g$ is the gravitational acceleration ($\mathrm{m\,s}^{-2}$), $P - E$ is precipitation minus evaporation ($\mathrm{m\,s}^{-1}$), $\mathbf{u}$ is the horizontal vector wind ($\mathrm{m\,s}^{-1}$), $q$ is specific humidity ($\mathrm{kg\,kg}^{-1}$), and $p$ is pressure (hPa). Overbars indicate monthly means and primes indicate daily mean departures from the monthly mean, which is transient eddy quantities, and the subscript $s$ denotes surface values. As such, the changes in $P - E$ is balanced by the moisture flux convergence by the mean flow (first term right-hand side) and by the transient eddy (second term right-hand side) and surface quantities. We denote $\delta(*) = (*)_{\mathrm{NMT/NGP}} - (*)_{\mathrm{CTL}}$ to represent the changes of the arbitrary quantity within parentheses between the

NMT or NGP and the CTL, thus Eq. (4) can be derived as:

$$\rho_w g \delta(P-E) = -\int_0^{P_s} \nabla \cdot \delta(\bar{u}\bar{q}) \mathrm{d}p - \int_0^{P_s} \nabla \cdot (\overline{u'q'}) \mathrm{d}p - \delta S \qquad (5)$$

The monthly mean moisture flux convergence changes can be further decomposed into dynamic contributions, which only involve the changes in $\boldsymbol{u}$ but no changes in $q$, and thermodynamic contributions, which only involve the changes in $q$ but no changes in $\boldsymbol{u}$. In addition, moisture flux convergence can be separated into the moisture advection and convergence of moisture flow. As such, Eq. (5) can be approximated as:

$$\rho_w g \delta(P-E) = -\int_0^{P_s} (\delta\bar{\boldsymbol{u}} \cdot \nabla \bar{q}_{CTL} + \bar{\boldsymbol{u}}_{CTL} \cdot \nabla \delta\bar{q} + \bar{q}_{CTL} \nabla \cdot \delta\bar{\boldsymbol{u}} + \delta\bar{q} \nabla \cdot \bar{\boldsymbol{u}}_{CTL}$$
$$+ \nabla \cdot \delta\overline{\boldsymbol{u}}\delta\bar{q}) \mathrm{d}p - \int_0^{P_s} \nabla \cdot \delta(\overline{u'q'}) \mathrm{d}p - \delta S \qquad (6)$$

The right-hand side of Eq. (6) includes, from left to right, column integrated mean flow moisture advection dynamic contribution, moisture advection thermodynamic contribution, convergence of moisture flow dynamic contribution, convergence of moisture flow thermodynamic contribution, nonlinear term (product of changes in both time mean flow and specific humidity), transient eddy contribution, and surface contribution. We refer refs. [54,55] for more details of moisture budget decomposition.

**EW responses to the Central American mountains**. The mountain-induced anomalous moist convection and diabatic heating over the ENP may also modulate the variability of tropical EWs that are locally initiated[32]. Since TCs observed in nature are often tightly associated with these disturbances of rich vorticity, it is also important to assess the EW responses to the Central American mountains in the numerical experiments. In this paper, EWs are detected and tracked using the algorithm adapted from the previous studies[56,57] that uses curvature vorticity anomalies at 700 hPa. To isolate the EWs from TCs, we removed all fields around detected TC centers by multiplying the weighting function before EW detections:[58,59]

$$w(x,y) = 1 - \exp\left[-\frac{r^2 \ln(4)}{2R^2}\right] \qquad (7)$$

where $r = r(x,y)$ is the distance of a grid point from the TC center and $R = 600$ km is the length scale of the weighting function. The weighting function is applied to a $10° \times 10°$ grid box centered on the TC centers at all time steps.

Supplementary Fig. 10a shows the track density of detected EWs in the CTL. EW track density is calculated by binning 6-hourly EW track locations every $2° \times 2°$ during each simulation season and spatially smoothed by 9-point moving average. Supplementary Fig. 10d, c indicate the anomalous EW track densities in the NMT and NGP simulations, respectively. It clearly suggests that the EW activities in the ENP are significantly enhanced (suppressed) in the NMT (NGP) simulations compared to the CTL, and these changes are primarily attributed to those locally generated EWs in the ENP rather than propagated from the Atlantic (Supplementary Fig. 10d–f).

## Data availability

All model output analyzed in the study have been deposited in the data servers at Texas A&M Supercomputing Facility in College Station, Texas, and can be made available from D.F. upon reasonable request. MSWEPv2.2 data are available at http://www.gloh2o.org/mswep/. J-OFURO3 data are available at https://j-ofuro.scc.u-tokai.ac.jp/en/. NOAA-OLR data are available at https://www.esrl.noaa.gov/psd/data/gridded/data.interp_OLR.html. CMIP6 data are available at https://esgf-node.llnl.gov/projects/cmip6/. All other data presented here are available at the Research Data Archive (https://rda.ucar.edu/) from the National Center for Atmospheric Research in Boulder, Colorado.

## Code availability

The source code of WRF model is available at https://www2.mmm.ucar.edu/wrf/users/downloads.html. TempestExtremes is available at https://github.com/ClimateGlobalChange/tempestextremes. The modified topography data used in the NMT and NGP simulations are available at https://zenodo.org/record/4894536 (https://doi.org/10.5281/zenodo.4894536).

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

## Acknowledgements

This work is a part of the lead author's (to D.F.) doctoral dissertation research directed by P.C., which was partially supported by the China Scholarship Council. The Texas Advanced Computing Center (TACC) at the University of Texas at Austin and the Texas A&M High-Performance Research Computing Facility provided computing resources for completing the simulation results reported in this paper. P.C. acknowledges support from the US National Science Foundation Grant AGS-1462127, the US Department of Energy under Award Number DE-SC0020072, and the US Department of Commence under Award Number NA20OAR4310409. C.M.P. acknowledges support from the US Department of Energy, Office of Science, Office of Biological and Environmental Research, Earth and Environmental Systems Sciences Division (EESSD), Regional and Global Model Analysis (RGMA) Program, under Award Number DE-AC02-05CH11231. This is a collaborative project between the Ocean University of China (OUC), Texas A&M University (TAMU), and the National Center for Atmospheric Research (NCAR), and completed through the International Laboratory for High-Resolution Earth System Prediction (iHESP), a collaboration between the Qingdao National Laboratory for Marine Science and Technology Development Center, Texas A&M University, and the National Center for Atmospheric Research.

## Author contributions

D.F. carried out the model simulations, analysis, and wrote the paper. P.C. conceived the original idea and provided interpretations of the physical mechanism. C.M.P. and R.S. contributed to interpreting the results. X.L. assisted with moisture budget analysis. H.E.B. provided the MSWEP data. All authors contributed to improving the paper.

## Competing interests

The authors declare no competing interests.
