## [Peer Review File · Nature Communications]

Reviewer comments–

Reviewer #1 (Remarks to the Author):

Overview

This study focuses on the role the Central American mountains play in tropical cyclone (TC) formation in the eastern North Pacific (ENP) basin on seasonal to climate timescales. Using the WRF regional climate model, the authors perform simulations with different terrain variations for the Central American mountains. They find that when the terrain is reduced to 50 m with land-use type similar to that of the Yucatan Peninsula, the net impact is to increase TC activity by 35% due to increased moisture transport into the ENP. This result provides a better understanding of the role the mountains play on synoptic to climate timescales, as well as the implications for using low resolution climate models with too heavily smoothed topography to simulate future TC activity.

Comments

This paper was previously submitted to Nature Geoscience and I served as a reviewer on that submission. The manuscript is largely unchanged from that submission, so most of my comments from my previous review still stand. Some of the minor edits did help clarify some points, though.

Overall, I feel that this paper is very well written. However, I still feel that the most important result of this study is that climate models used to simulate future TC activity have low resolution with too heavily smoothed topography. Thus, they would not accurately represent the TC activity in the ENP, which also has downstream impacts. This point should be emphasized earlier and not just in the final paragraph of the conclusion. Additional discussion of the heavily smoothed terrain in these climate models as part of the reason the authors chose to simulate the terrain in different ways for their study is needed.

Instead, the paper focuses on how the presence of the Central American mountains actually inhibits TC activity on seasonal to climate timescales by blocking moisture transport into the ENP. While this is a very interesting finding and is helpful for gaining a better understanding of the role the mountains play in ENP TC activity, even on synoptic timescales, I think that the implications for the climate models are more impactful. There is also no discussion on whether the climatological values of moisture in the ENP are sufficient without any additional moisture transport, to support cyclogenesis.

In addition, there is very little analysis on how the removal of the mountains impacts the propagation of easterly waves, or other sources of vorticity, from the Atlantic into the Pacific. Without this analysis, the authors may be falsely attributing the increased TC activity to increased moisture transport rather than increased easterly wave propagation, or a combination of the two, into the basin.

The methods for testing the role of the Central American mountains seem robust. I am not an expert when it comes to climate models, but from what experience I do have, it appears that the authors have been thorough. Performing the simulations at 27 km and 9 km resolution and obtaining similar results improves the confidence in their findings, as using a coarse resolution is not ideal for simulating TCs. The statistical analysis performed also supports their conclusions.

I would find it helpful to include maps of TC genesis density and track density from observations. This would serve as additional support for the validity of your control simulation and provide the reader with additional confidence in your results. You do this in your supplementary figures for precipitation, 10-m wind, latent heat flux, and outgoing longwave radiation to validate the datasets you use, so it would be nice to see something similar for the model.

Reviewer #2 (Remarks to the Author):

See next page -

This paper examines the impact of Central American mountains on tropical cyclone activity, using a suite of simulations with a regional weather model. The authors find that Central American mountains reduce tropical cyclone activity by blocking moisture transport from the Atlantic, which in turn reduces convective activity and mid-tropospheric relative humidity, a key driver of tropical cyclone genesis. Analogous results are found when closing the gaps in the Central American mountains. The results are shown to be robust via simulations at 9 km and 27 km resolutions, and supporting analysis of background climate fields.

Overall, I find this an impressively well-thought-out and interesting study. I think it contributes significantly to understanding of the controls on tropical cyclone activity in the eastern North Pacific, and importantly helps clarify the differences in the role of this topography at weather vs climate time scales, which has been a confusing point in the literature. Additionally, the paper is for the most part quite clearly written, with helpful figures. I think the general contents of this paper are very appropriate for Nature Communications.

However, I have a few larger critiques/questions which I would like to see addressed listed immediately below, and then line by line suggestions for how to improve the paper listed farther below.

- 1) The work is motivated by talking about how the ENP tropical cyclone genesis density is the highest on the planet. However, the conclusion is that mountains actually decrease TC activity in this region. If mountains are not driving the anomalously high TC genesis in this region, what do you think is and how can the community move forward in understanding this region? I suggest exploring this question a bit in the discussion.
- 2) I think it should be clarified earlier in the text that the models used are not atmosphere-ocean coupled, and that this is a limitation of this study. For someone unfamiliar with the particular version of WHARF that might not be immediately clear. I think this should be added in addition to the existing text in the Discussion/Conclusion on this point.
- 3) In [183-228] the authors argue that the Atlantic-Pacific integrated water vapor transport is driving the changes in TCs and precipitation. While I agree with this conclusion, and find Figure 4 compelling, I am a bit confused by Figure 3, and the related explanations. I would think the role of this Atlantic-Pacific water vapor transport could be more cleanly diagnosed using a moisture budget decomposition (ala Seager and Vecchi 2010 or Baldwin and Vecchi 2016 links copied below), to decompose the role of changes in humidity versus winds, at monthly mean vs transient time scales. Right now, the plot showing divergent wind does not seem very relevant to me, because it is $u \cdot q$ not u by itself that matters for changes in precipitation/ convection. To me, the plots showing changes in buoyancy and convection are helpful, but still not fully addressing the question of where the moisture is coming from, as far as I understand right now. I would appreciate it if the authors could help me understand their logic a bit more here, to square with how I would normally think about this problem, or try the alternative approach I suggest of a moisture budget decomposition and see if it is helpful.

Greenhouse warming and the 21st century hydroclimate of southwestern North America

R Seager, GA Vecchi - ... of the National Academy of Sciences, **2010** - National Acad Sciences

Influence of the Tian Shan on arid extratropical Asia

J Baldwin, G Vecchi - Journal of Climate, **2016** - journals.ametsoc.org

- 4) While the central figures of the paper, and most of the supplemental figures are directly relevant to this work, the discussion of model resolution and precipitation (and related figures SI Fig. 13 and 14) feel like they are the beginnings of an entirely different paper. I understand the connection to the current work, and find the results of this analysis compelling, but I'm wondering if it would be better for the scientific community if this was a separate paper. I also have mixed feelings about the current structure of introducing these new results in the final paragraph of the paper. If you definitely want to include them, could you introduce them in the results section, and then discuss them further in the Discussion section?
- 5) In [234-235] you assert that "the anomalous deep convection and heating also modulate the occurrence of the easterly waves". While you show the changes in easterly waves, I am not sure how you can conclude that this is due to the change in deep convection and heating. Can you please back this up with either reference to relevant literature or some further explanation?
- 6) [362-374] I find this description of how the gaps are closed rather confusing. First, what is meant by "At each meridional grid within the box"? Second, does a random factor of order 1 mean between 1 and 9? Finally, can you explain more clearly why you use this somewhat complicated method with some randomization, as it's not immediately intuitive to me? Finally, is the method (including the Gaussian filter of 5 zonal and 3 meridional gridcells) same for the 9 km simulations?
- 7) [517-521] The authors say they will make the code used in this study available upon request. I want to ensure that this is consistent with Nat Comms data policies—it would be best if this could be put on github or the like to be openly accessible.

Line by line comments:

- 1) [13] suggest to say "tropical cyclones (TCs) on earth"
- 2) [18] suggest to remove "," after "show"
- 3) [23] "mechanism" should be "mechanisms"
- 4) [32] suggest to remove "the" before "TC activity density"
- 5) [47] suggest to move citations 12,13 to end of the sentence
- 6) [54] should be "take into consideration the unique"
- 7) [55-56] suggest to write just "Sierra Madre in North America" as I've seen varying interpretations of exactly what "Sierra Madre Occidental" refers to
- 8) [56] suggest to say "~1km high"
- 9) [57] suggest to replace "marked" with "interrupted" or "split-up"
- 10) [63] suggest to remove "-" between "mountain" and "gaps"
- 11) [82] should be "model"

- 12) [103] should mention briefly here why focused on Tehuantepec and Papagayo but left the Panama gap as is
- 13) [137-138] I don't understand what is meant by "Combining the opposite sign of changes together"
- 14) [140] should be "mountains as well"
- 15) [142-143] Does the removal of these jets also explain the nmtn decrease in TC activity in this region?
- 16) [154] You say "mountains can fuel TC genesis". Is this backed up by the literature? Or would it be more accurate to say "mountain gaps can fuel TC genesis"?
- 17) [156] suggest to add "near" before "identical"
- 18) [157] suggest to replace "it provides" with "there is"
- 19) [165-166] suggest to move "well" before "represents"
- 20) [172] "terms" should be "term"
- 21) [179] suggest to move second apostrophe around the two pseudo phrases before the parentheses.
- 22) [188] suggest to replace "which is directly resulting" with "which directly results"
- 23) [193] suggest to change "topographic precipitation" to "orographic precipitation"
- 24) [209] should say "resulting in more precipitation"
- 25) [229] suggest to remove "In particular"
- 26) [230] I think this would read clearer if you said "Removal of the mountains' blocking effect"
- 27) [230] "diabetic" should be "diabatic" (here and anywhere else—I think I saw this one other place)
- 28) [232] suggest to replace "it" with "this teleconnection"
- 29) [295] "seasons" should be "season"
- 30) [304] What region is the nest performed over? Can you specify that here for clarity?
- 31) [311] suggest to remove "We specially note that"
- 32) [313] What is meant by "convexity" here? I'm not used to seeing this term describing topography.
- 33) [318] Can you briefly say how these relative vorticity thresholds are chosen?
- 34) [322] "topography height" should be "topographic height"
- 35) [338] suggest to add "with observations" after "precipitation"
- 36) [338-352] Are all these biases generally consistent for both model resolutions? The comparisons shown in the SI are just for the 27 km model, right?
- 37) [345-346] suggest to change to "This bias in South American orographic precipitation"
- 38) [348-349] suggest to "CCMPv2 and CTL shows significant consistency"
- 39) [359] should be "Papagayo"
- 40) [368] suggest to change to "a size of 5 zonal and 3 meridional gridcells and a standard deviation..."
- 41) [396] "purposed" should be "proposed"
- 42) [400] suggest to remove "the" before "climate change"
- 43) [402] suggest to add "the" before "vertical and horizontal wind speeds"
- 44) [409] remove "that" before "determined by"

- 45) [416] shouldn't this be "determined from" not "defined by"? since you provide the actual definition on the next page
- 46) [425] should have "is" before "determined"
- 47) [433] based on the prior line, shouldn't there be a factor of 2 in front of $(h_{10m} - h^*_{env})$?
- 48) [440] "diabetic" should be "diabatic"
- 49) [448] suggest to write "; as such, we need"
- 50) [449] add "such" before TCs
- 51) [452] do you mean "that are adapted"?
- 52) [456] add "the" before "95%"
- 53) [492,498] "smoothened" should be "smoothed"
- 54) [501] suggest to change to "similar to most subseasonal-to-seasonal prediction models; as such, the results"
- 55) [Fig 1 boxplots] What lat-lon region is the averaging performed over? Please specify in caption.
- 56) [678] here and elsewhere you write "denoated". It should be "denoted"
- 57) [679] "experiment" should be "experiments"
- 58) [683,699] suggest to add "the" before "two-sided t-test"
- 59) [Fig 2] Why do you use a different interval for the model vs ibtracs TC genesis density? Isn't this a bit misleading?
- 60) [719] should be "Sensitivity"
- 61) [724] should be "quadrant"
- 62) [727] should be "activity"
- 63) [733] "minus" should be "subtracting"
- 64) [Fig 4a] Do the seasonal ensemble members for the CTL fall along this line too? If not, why not? I wasn't sure why that wasn't included in this figure.
- 65) [Fig 3] Is the black stuff at the bottom topography? Please explain what this is in the caption and how it is determined (ie meridional max, average, etc)
- 66) [Fig S5] In the caption, you note panel c) but there is no panel c). Do you mean a or b?
- 67) [Fig S9] "neglectable" should be "negligible"
- 68) [Fig S10] is "changes" in initial bolded line a mistake?
- 69) [Fig S11] should be "distribution"
- 70) [Fig S12] should be "Easterly waves"
- 71) [Fig S13] How do you construct the high res and low res topography fields in panels g and h? Do you average the topography across models?
- 72) [Fig S13] should say "purposes, precipitation on various"
- 73) [Fig S13] should say "Refer to Methods"
- 74) [Fig S14] What does TCM stand for? Also this is also a June-Nov average too, right? Should maybe specify this.

Reviewer 3 –

Next page -

Title: Central American Mountains Inhibit Eastern North Pacific Seasonal Tropical Cyclone Activity

Authors: Fu, Chang, Patricola, Saravanan, and Beck

Overall Recommendation: Major Revisions

Summary: This manuscript presents the interesting finding that tropical cyclone activity in the east north Pacific (ENP) is negatively affected by the topography of Central America, which is counter to some previous case studies. A novel aspect of this study is the use of a large climatological simulation period with a similarly large set of ensembles. Relatively coarse (27-km) and high resolution (9-km) simulations were carried out to show results were insensitive to resolution, though the higher resolution simulations are only included in supplementary material. A possible mechanism (moisture transport leading to increased buoyancy and ascent) is hypothesized for why removal of Central American terrain leads to a significant increase in TC activity in the ENP. There are a number of high quality figures that help illustrate the key results of this interesting manuscript.

However, I do have reservations about the structure of this manuscript. There are a large number of supplemental figures (14) which is more than three times than the total figures included in the main manuscript text (4). Moreover, these supplemental figures are referred to in both the results and methodology of this manuscript and seem more integral to the manuscript than just being included as supplemental material. This reviewer would like to see substantial revision of the manuscript, incorporating some of the more integral supplemental figures provided into the primary manuscript. There are also a few extraneous topics included in the methodology that seem only tangential to the main theme of the manuscript (for example: how easterly waves and precipitation are affected by differences in topography) that this reviewer would prefer to see incorporated into an additional manuscript.

Therefore I am recommending major revisions to this manuscript before I can consider this manuscript suitable for publication. My comments sorted by major and specific are below:

Major Comments:

- 1) There is a lot of extraneous material in the methodology section that does not seem applicable to the manuscript theme which focuses on how Central American mountains can impact TC activity in the ENP. While the first four sections of the methods (High-resolution climate model simulations, Experiment designs, Modified genesis potential index, Buoyancy diagnostics) are all relevant to the manuscript, the final two sections (Easterly wave tracking and CMIP6 Precipitation; Lines 439-504) are too far removed from the manuscript theme to add overall value to this manuscript and seem like unnecessary extra material.

While it is true that easterly waves and precipitation are both meteorological phenomena that can influence ENP TC activity, the detailed description of the methods and results presented in the supplemental figures go beyond the scope of what this manuscript was inferred to discuss in the introduction (line 73-77).

The recommendation of this reviewer would be to remove this extraneous material that is already supplemental as organized by the authors. Specifically lines 439-504 and supplementary figures 12–14. These results and figures are high quality but would be more suitable for a subsequent manuscript.

- 2) There are a number of instances where this reviewer would have preferred to structurally include the supplemental figures as regular figures in the text. For example, Supplemental figure 1 is a nice introductory figure that is the only figure discussed in the introduction. Why it is considered supplementary when it provides important background information provided in lines 52-67? There are a number of instances where it would have been preferable to include supplemental figures as regular figures in the manuscript and this will be occasionally described in the specific comments below.

Specific Comments:

L43: Instead of “tightly” use “closely” instead?

L41-51: A few additional citations worth including in this portion of the introduction is the dramatic influence of convectively coupled Kelvin waves which have also been shown to have a significant influence on tropical cyclone development in the ENP. See citations below:

Schreck, C. J., 2016: Convectively Coupled Kelvin Waves and Tropical Cyclogenesis in a Semi-Lagrangian Framework. *Mon. Wea. Rev.*, 144, 4131-4139, <https://doi.org/10.1175/MWR-D-16-0237.1>.

Camargo, S. J., and Coauthors, 2019: Tropical Cyclone Prediction on Subseasonal Time-Scales. *Trop. Cyclone Res. Rev.*, 8, 150–165, <https://doi.org/10.1016/j.tcr.2019.10.004>.

L82: “model” not “mode”

L105-107: This statement seems out of place with the rest of the paragraph. It might fit in better after the end of the sentence on line 85. Also can there be a quick explanation on why a tropical channel domain is better to handle mountain remote influences on TCs?

L108-117: This reviewer very much appreciates that the 27-km simulation results are being compared to the 9-km simulations described in this portion of the results. But I would have preferred to see more details comparing the two simulations in the following section (lines 146-153) especially given the knowledge higher resolution would be more likely to capture important mesoscale circulations produced by the terrain gaps and have better resolution to capture TC circulation and wind field aspects not touched on in this manuscript.

L141-142: Use “Gulf of Tehuantepec” instead of just Tehuantepec?

L142-143: Also reference supplemental figure 1 here since you are referring to the jet curvature seen in Figs. 1 and 4? This is another instance where the supplemental figures would be better off being in the primary manuscript.

L172-178: This passage nicely explains how GPI is broken down into its individual components but is another example where the supplemental figure 7 could have been moved to right after Fig. 2 in the primary manuscript. Instead the reader is forced to hunt town this important figure for this passage in another location. Structurally this makes the manuscript more difficult to read.

L180 and in other places: “state-of-art” is superfluous.

L204-214: Missing in this TC genesis link here is how enhanced buoyancy and moisture results in vorticity spin up necessary for tropical cyclogenesis. I agree with the authors that higher vertical velocity and buoyancy via enhanced moisture is likely to result in more favorable conditions for tropical cyclogenesis but need to also describe how stronger deep moist ascent also results in vortex stretching and aggregation of pre-existing larger-scale vorticity. See citations below:

Davis, C., C. Snyder, and A. C. Didlake, 2008: A Vortex-Based Perspective of Eastern Pacific Tropical Cyclone Formation. *Mon. Weather Rev.*, **136**, 2461–2477, <https://doi.org/10.1175/2007MWR2317.1>.

Wang, Z., 2014: Role of Cumulus Congestus in Tropical Cyclone Formation in a High-Resolution Numerical Model Simulation. *J. Atmos. Sci.*, **71**, 1681–1700, <https://doi.org/10.1175/JAS-D-13-0257.1>.

L230: Diabatic... not diabetic

L336: Is this using the CTL simulation in the 27-km domain or the 9-km domain? Also this information seems less about the methods, but more about verifying the control simulation with reanalysis and observation datasets. This would have made for a nice section in the results before getting into the main simulation comparisons using different terrain types. This comment could also be applied to the discussion related to Supplemental Figures 4-6.

L397: Again state-of-art is superfluous and not necessary here.

L439-504: This portion of the methods section seems unnecessarily to the rest of the manuscript. How the CTL simulation depicts easterly waves and precipitation patterns is likely an important discussion but seems beyond the scope of this study that already includes 18 figures (14 supplementary). Recommend cutting this portion of the manuscript as discussed in major comment #1.

Revision Note NCOMMS-20-31758

“Central American mountains inhibit eastern North Pacific seasonal tropical cyclone activity” by Dan Fu, Ping Chang, Christina M. Patricola, R. Saravanan, **Xue Liu**, and Hylke E. Beck.

During the revision process, Xue Liu (Department of Oceanography, Texas A&M University, College Station, Texas, USA) has provided considerable assistance in the moisture budget and moisture budget decomposition, as well as editing the manuscript. Therefore, we added her name to the co-authors' list.

Point-to-Point Reply to Referee #1

First, we would like to thank the referee for the invaluable comments and help us improve the manuscript. We have carefully followed each of your comments listed in red and revised our paper accordingly. Our replies to your comments are as follows:

Referee #1: This paper was previously submitted to Nature Geoscience and I served as a reviewer on that submission. The manuscript is largely unchanged from that submission, so most of my comments from my previous review still stand. Some of the minor edits did help clarify some points, though.

Overall, I feel that this paper is very well written. However, I still feel that the most important result of this study is that climate models used to simulate future TC activity have low resolution with too heavily smoothed topography. Thus, they would not accurately represent the TC activity in the ENP, which also has downstream impacts. This point should be emphasized earlier and not just in the final paragraph of the conclusion. Additional discussion of the heavily smoothed terrain in these climate models as part of the reason the authors chose to simulate the terrain in different ways for their study is needed.

Reply: Thank you for taking the time to serve as a referee of our work for the second time. Your constructive comments and positive feedbacks are highly appreciated. We agree with you that this paper may have important implications on topography representation used in climate models, as such; we add some discussions in the

Introduction. Please see lines 72-82. However, as criticized by Referee #2 and #3 about manuscript structure, in the revision, we removed the topic of how smoothed topography could influence precipitation in the CMIP6 (Supplementary Figure 13 and 14 of the initial submission). We will expand the results presented in Supplementary Figure 13 and 14 and make more comprehensive analyses in a separate future study.

Referee #1: Instead, the paper focuses on how the presence of the Central American mountains actually inhibits TC activity on seasonal to climate timescales by blocking moisture transport into the ENP. While this is a very interesting finding and is helpful for gaining a better understanding of the role the mountains play in ENP TC activity, even on synoptic timescales, I think that the implications for the climate models are more impactful. There is also no discussion on whether the climatological values of moisture in the ENP are sufficient without any additional moisture transport, to support cyclogenesis.

Reply: We agree this paper has significant implications for climate models' topography representation; see our answer to this point above. The figure below compares the simulated column integrated water vapor (IWV) in CTL with the satellite observations. It shows that CTL yields a realistic climatological value of moisture in the eastern North Pacific (ENP), particularly in the area with strongest TC activity changes (i.e. 110°W-100°W, 10°N-15°N). The moisture value in the Panama Bight is somewhat underestimated in CTL. This general agreement between the simulated and observed climatological moisture in ENP gives us the confidence that the model bias in mean moisture in ENP does not impact the robustness of our findings. Regarding your question “whether the climatological values of moisture in the ENP are sufficient without any additional moisture transport, to support cyclogenesis”, as can be seen from Figure 1 in the revision, the strong moisture transport from Atlantic to the ENP is persistent on the seasonal timescale, thus it's difficult to consider a scenario that ENP climatological moisture is not influence by this moisture transport from Atlantic. Comparing our CTL with ERA5 reanalysis, ENP 700hPa relative humidity is also reasonably simulated, and the climatological

value is comparable to those in the western North Pacific warm pool region. Note that, western North Pacific has the strongest TC activity over the globe that primarily attributed to the warm sea surface temperature and abundant moisture supply. All these results manifest that CTL simulated moisture in the ENP is faithful and the climatological values are sufficient for TC genesis.

Figure R1-1: Satellite retrieved and CTL simulated June-November averaged column integrated water vapor (IWV; unit: mm). The black boxes indicate the area of 110°W-100°W, 10°N-15°N.

Figure R1-2: ERA5 and CTL simulated June-November averaged 700 hPa relative humidity (unit: %). Black contours indicate the area with 70, 75 and 80% relative humidity values.

Referee #1: In addition, there is very little analysis on how the removal of the mountains impacts the propagation of easterly waves, or other sources of vorticity, from the Atlantic into the Pacific. Without this analysis, the authors may be falsely attributing the increased TC activity to increased moisture transport rather than increased easterly wave propagation, or a combination of the two, into the basin.

Reply: In the revised manuscript, we changed the easterly wave detection algorithm from Eulerian to Lagrangian method based on Belanger et al. (2016) and Whitaker and Maloney (2018). Specifically, we detected and tracked easterly wave trough associated curvature vorticity anomalies, and divided the easterly waves in the ENP into two groups based on their origins, that is, whether they are generated locally in the ENP or propagated from Atlantic. The figure below (also shown as Supplementary Figure 10 in the revised manuscript) shows that the changes of ENP easterly waves in NMT and NGP are primarily attributed to those locally generated easterly waves, which are caused by the anomalous moisture transport induced deep moisture convection and diabatic heating.

Figure R1-3: a) The ensemble mean of seasonal easterly wave track density (occurrence number decade⁻¹) in CTL and the differences of b) NMT minus CTL and c) NGP minus CTL. d-f) Same as a-c), except for those easterly waves originated in the ENP (not propagated from the Atlantic). Grey dots denote statistical confidence at the 95% level based on the two-sided t-test.

Belanger, J. I., M. T. Jelinek, and J. A. Curry, 2016: A climatology of easterly waves in the tropical Western Hemisphere. *Geosci. Data J.*, **3**, 40-49, <https://doi.org/10.1002/gdj3.40>.

Whitaker, J. W., and E. D. Maloney, 2018: Influence of the Madden–Julian Oscillation and Caribbean low-level jet on east Pacific easterly wave dynamics. *J. Atmos. Sci.*, **75**, 1121–1141, <https://doi.org/10.1175/JAS-D-17-0250.1>.

Referee #1: The methods for testing the role of the Central American mountains seem robust. I am not an expert when it comes to climate models, but from what experience I do have, it appears that the authors have been thorough. Performing the simulations at 27 km and 9 km resolution and obtaining similar results improves the confidence in their findings, as using a coarse resolution is not ideal for simulating TCs. The statistical analysis performed also supports their conclusions.

I would find it helpful to include maps of TC genesis density and track density from observations. This would serve as additional support for the validity of your control simulation and provide the reader with additional confidence in your results. You do this in your supplementary figures for precipitation, 10-m wind, latent heat flux, and outgoing longwave radiation to validate the datasets you use, so it would be nice to see something similar for the model.

Reply: We thank you for these constructive comments. The simulated and observed TC genesis densities are shown in Figure 3a and 3d in black contours with marked label levels. We now added TC track density maps to Supplementary Figure 1e and 1j as color shades.

Point-to-Point Reply to Referee #2

We would like to thank the Referee for taking the time to review our manuscript and offering helpful suggestions to improve it. We are particularly grateful for your suggestions on the moisture budget decomposition, which led to a very nice addition to the manuscript. We appreciate your constructive comments and have carefully considered each of these comments listed in red and revised our paper accordingly. Our replies to your comments are as follows:

Referee #2: Overall, I find this an impressively well-thought-out and interesting study. I think it contributes significantly to understanding of the controls on tropical cyclone activity in the eastern North Pacific, and importantly helps clarify the differences in the role of this topography at weather vs climate time scales, which has been a confusing point in the literature. Additionally, the paper is for the most part quite clearly written, with helpful figures. I think the general contents of this paper are very appropriate for Nature Communications.

Reply: We appreciate these positive comments and are very happy that you find this paper a useful contribution to the understanding of the controls on TCs in the eastern North Pacific (ENP).

However, I have a few larger critiques/questions which I would like to see addressed listed immediately below, and then line by line suggestions for how to improve the paper listed farther below.

Referee #2: The work is motivated by talking about how the ENP tropical cyclone genesis density is the highest on the planet. However, the conclusion is that mountains actually decrease TC activity in this region. If mountains are not driving the anomalously high TC genesis in this region, what do you think is and how can the community move forward in understanding this region? I suggest exploring this question a bit in the discussion.

Reply: Besides the very unique contributions from the Central American mountains, as we noted in the Introduction, ENP TC genesis is also influenced by multiple factors.

On the synoptic scale, observations reveal that the majority of TCs in the ENP are directly triggered by or closely associated with tropical easterly waves, convectively coupled Kelvin waves, breakdown of the intertropical convergence zone (ITCZ), mesoscale convective systems, or the monsoon trough, through which lower-tropospheric cyclonic vorticity is maximized. In particular, the Pacific ITCZ stays in the ENP during the JJASON, which contributes to lots of convective tropical disturbances. Under sufficient favorable environmental conditions, such as warm SSTs, weak vertical wind shear and humid mid-level moisture, these tropical disturbances can eventually develop to TCs. Although quantitatively it is hard to pinpoint which factor contributes the most, combining all these factors together, the ENP has the highest density of TCs in terms of climatology.

To better understand TC activities in the ENP, we think high-resolution numerical simulations are essential. On one hand, high-resolution simulations can better capture the dynamic/thermodynamic processes that trigger TC genesis and also the internal modes of climate variability that can modulate the large-scale environmental TC favorability; on the other hand, high-resolution simulations are ideal to bridge the gap between weather and climate and help us better disentangle the complex interactions between the synoptic TCs and the large-scale environments that drive them.

In the revised manuscript, we explored this question in the discussion section. Please see lines 288-308.

Referee #2: I think it should be clarified earlier in the text that the models used are not atmosphere-ocean coupled, and that this is a limitation of this study. For someone unfamiliar with the particular version of WHARF that might not be immediately clear. I think this should be added in addition to the existing text in the Discussion/Conclusion on this point.

Reply: Thanks for point this. We noted this limitation before showing the main conclusion in the revised manuscript. Please see lines 94-101.

Referee #2: In [183-228] the authors argue that the Atlantic-Pacific integrated water vapor transport is driving the changes in TCs and precipitation. While I agree with this conclusion, and find Figure 4 compelling, I am a bit confused by Figure 3, and the related explanations. I would think the role of this Atlantic-Pacific water vapor transport could be more cleanly diagnosed using a moisture budget decomposition (ala Seager and Vecchi 2010 or Baldwin and Vecchi 2016 links copied below), to decompose the role of changes in humidity versus winds, at monthly mean vs transient time scales. Right now, the plot showing divergent wind does not seem very relevant to me, because it is $u \cdot q$ not u by itself that matters for changes in precipitation/ convection. To me, the plots showing changes in buoyancy and convection are helpful, but still not fully addressing the question of where the moisture is coming from, as far as I understand right now. I would appreciate it if the authors could help me understand their logic a bit more here, to square with how I would normally think about this problem, or try the alternative approach I suggest of a moisture budget decomposition and see if it is helpful.

Reply: Thank you for suggesting these nice papers. We removed the divergent wind analysis in the revised manuscript. Following your suggestion, we made a moisture budget analysis and decomposed the moisture flux convergence changes into mean flow (monthly mean) and transient eddy (daily mean departures from monthly mean) component, and further into dynamic or thermodynamic contributions. Please see the attached two figures that are now included in the Supplementary Information as Supplementary Figure 5 and 6. The detailed moisture budget analysis and decomposition are described in the Methods.

In CTL, it clearly shows that precipitation minus evaporation ($P-E$) is primarily controlled by the mean flow (monthly mean) moisture flux convergence, while the moisture flux convergence by transient eddy plays a minor but negative role, which indicates that high-frequency synoptic variability tends to transport the moisture from equator to mid-latitude in the ENP. Surface term largely induces positive moisture

flux convergence, but its contribution to the total column integrated moisture is negligibly small.

Figure R2-1: Moisture budget terms from CTL simulation ensemble mean. a) Precipitation minus evaporation ($P-E$; mm day^{-1}), b) moisture flux convergence by the monthly mean flow (mm day^{-1}), c) moisture flux convergence by the transient eddy (daily mean departures from the monthly mean; mm day^{-1}) and d) surface quantities (mm day^{-1} ; note that the colorbar scale is different) term. Refer the Methods section for the details of moisture budget equation and decomposition.

Figure R2-2a,b (R2-2d,e) further show NMT (NGP) mean flow and transient eddy moisture flux convergence differences from CTL. As shown in Figure 5 of revised manuscript (also indicated as black contours in the figures below), the enhanced (reduced) $P-E$ in NMT (NGP) is due to mean flow changes, rather than transient eddies. Again, transient eddies tend to offset the changes by mean flow. By decomposing the mean flow moisture flux convergence $[-\int_0^{P_s} \nabla \cdot (\bar{\mathbf{u}}\bar{q}) dp]$ into moisture advection $(-\int_0^{P_s} \bar{\mathbf{u}} \cdot \nabla \bar{q} dp)$ and convergence of moisture flow $(-\int_0^{P_s} \bar{q} \nabla \cdot \bar{\mathbf{u}} dp)$ to further determine the relative roles of thermodynamic and dynamic contributions played in NMT (NGP) moisture budget change (Figure R2-2c,f), we found that it is the dynamic contribution of moisture flow convergence

$[-\int_0^{P_s} (\bar{q}_{CTL} \nabla \cdot \delta \bar{\mathbf{u}}) dp]$ dominates net $P-E$ changes in the area of 120°W to 93°W , 8°N to 15°N . It suggests that the changes of moisture in the ENP are primarily driven by circulation changes, rather than the local moisture changes (the consequence of atmospheric warming).

Figure R2-2: Moisture budget decomposition. The differences of moisture flux convergence a) by the mean flow (mm day^{-1} ; colorshadings) and b) by the transient eddy (mm day^{-1} ; colorshadings) using NMT minus CTL ensemble mean. The precipitation minus evaporation difference ($P-E$; mm day^{-1}) is overlaid in black contours (contour starting from -2.5 with the interval of 1 ; positive solid lines and negative dotted lines). c) Relative contributions of various moisture budget terms to changes in hydrological cycle averaged over the red box (120°W to 93°W , 8°N to 15°N) in panel a). Refer to Methods for details of moisture budget analysis and decomposition of moisture flux convergence. d-f) Similar to a-c), but for the NGP minus CTL. Grey dots denote statistical confidence at the 95% level based on the two-sided t-test.

To better illustrate this result, we plot the mean circulation change in the figure below (now included as Supplementary Figure 4 in the revised manuscript). It shows that the low-level winds from the Atlantic can extend further (less further) into the ENP in

NMT (NGP) as the Central American mountains are removed (the mountain gaps are closed), and thus can increase (decrease) low-level wind convergence in the ENP (Figure R2-3g,h; Figure R2-3m,n). However, we note that the significant increase (decrease) of flow convergence only confines to the narrow area in the eastern portion of ENP close to the coastline (Figure R2-3b,h,n), whereas the increase (decrease) of moisture flux (qu) convergence occurs at much broader area (Figure R2-2a,d). It suggests that, in NMT (NGP), the increased (decreased) moisture flux from the Caribbean (please see Figure 1) enhances (reduces) the ENP moisture content (Figure R2-3e,k,q), also the buoyancy, primarily due to stronger (weaker) moisture flow convergence, and thus increase (decrease) moisture convection there (Figure R2-3f,l,r). Although other terms contribute differently in NMT and NGP in the moisture budget, dynamic contribution of moisture flow convergence clearly dominates the budget. We added this part of analysis in the revised manuscript. Please see lines 240-258.

Figure R2-3: Topography-induced responses of mean circulation. Seasonal averaged a) 925hPa wind speed (m s^{-1}), b) 925hPa horizontal wind divergence (10^{-5} s^{-1}), c) 925hPa relative vorticity (10^{-5} s^{-1}), d) 925 hPa vorticity equation vortex stretching term $[-(f + \zeta)(\partial u/\partial y + \partial v/\partial x); 10^{-10} \text{ s}^{-2}]$, e) column integrated water vapor (IWV; mm), and f) 500 hPa vertical wind (cm s^{-1}) from CTL simulation ensemble mean. g-l) And m-r) are similar, but from the differences using the NMT minus CTL and NGP minus CTL, respectively. Grey dots denote statistical confidence at the 95% level based on the two-sided t-test. Yellow sketchings indicate the mountain outlines at 925 hPa.

Referee #2: While the central figures of the paper, and most of the supplemental figures are directly relevant to this work, the discussion of model resolution and precipitation (and related figures SI Fig. 13 and 14) feel like they are the beginnings of an entirely different paper. I understand the connection to the current work, and find the results of this analysis compelling, but I'm wondering if it would be better for the scientific community if this was a separate paper. I also have mixed feelings about the current structure of introducing these new results in the final paragraph of the paper. If you definitely want to include them, could you introduce them in the results section, and then discuss them further in the Discussion section?

Reply: In the revised manuscript, we deleted this portion. We will publish this portion of the analysis in a separate future paper.

Referee #2: In [234-235] you assert that “the anomalous deep convection and heating also modulate the occurrence of the easterly waves”. While you show the changes in easterly waves, I am not sure how you can conclude that this is due to the change in deep convection and heating. Can you please back this up with either reference to relevant literature or some further explanation?

Reply: Rydbeck and Maloney (2014) investigated the perturbation available potential energy (PAPE) and perturbation kinetic energy (PKE) budgets of easterly wave composites and found the generation of PAPE associated with perturbation diabatic heating that is subsequently converted to PKE is the dominant energy source for easterly waves during the MJO neutral and active phases in the ENP. Specifically, the convergence increase at mid-levels resulting from enhanced diabatic heating can intensify the background cyclonic vorticity, and thus induce increased barotropic conversion and more active easterly waves. We added this reference in the revised manuscript.

Rydbeck, A. V., & Maloney, E. D., Energetics of East Pacific easterly waves during intraseasonal events, *J. Clim.*, **27**, 7603–7621 (2014).

Referee #2: [362-374] I find this description of how the gaps are closed rather confusing. First, what is meant by “At each meridional grid within the box”? Second, does a random factor of order 1 mean between 1 and 9? Finally, can you explain more clearly why you use this somewhat complicated method with some randomization, as it’s not immediately intuitive to me? Finally, is the method (including the Gaussian filter of 5 zonal and 3 meridional gridcells) same for the 9 km simulations?

Reply: We apologize for the confusing description. “At each meridional grid within the box” means “At each grid point (longitude, latitude) within the box”. “a random factor of order 1” should be “a random factor in the range between 1 and 1.05”. So we have rewritten the description as follows: “At each grid point within the box, we first defined the latitude-dependent critical values as box zonal mean plus one standard deviation of the zonal values, then the elevations lower than the latitude-dependent critical values were multiplied by a random factor in the range between 1 and 1.05 repeatedly, until the modified elevations were larger than the critical values.” The actual earth surface elevation is not flat and it always has some deviations even in a small region. In our NGP experiments, we want to treat our modified topographic height to have some realistic variations, rather than an unrealistic flat surface. This is the reason that we employed such a relatively complex method. For the 9-km simulations, we used 15 zonal and 9 meridional grid cells as the window of Gaussian filter. We have noted this in the revised manuscript. Please refer lines 397-409.

Referee #2: [517-521] The authors say they will make the code used in this study available upon request. I want to ensure that this is consistent with Nat Comms data policies—it would be best if this could be put on github or the like to be openly accessible.

Reply: As the simulations produce more than 100TB of output files, we currently don't have appropriate online storage like GitHub to directly share the model outputs and make it public available. However, we uploaded the modified topography data used in the simulations to Github (<https://github.com/fudan1991/>

Tropical-Channel-Model), and would like to share the data upon request to D. F. (fudan1991@tamu.edu).

Line by line comments:

[13] suggest to say “tropical cyclones (TCs) on earth”

Reply: Done.

[18] suggest to remove “,” after “show”

Reply: Done.

[23] “mechanism” should be “mechanisms”

Reply: The change has been made.

[32] suggest to remove “the” before “TC activity density”

Reply: Done.

[47] suggest to move citations 12,13 to end of the sentence

Reply: Done.

[54] should be “take into consideration the unique”

Reply: Done.

[55-56] suggest to write just “Sierra Madre in North America” as I’ve seen varying interpretations of exactly what “Sierra Madre Occidental” refers to

Reply: Thanks for pointing this. We corrected it in the revised manuscript.

[56] suggest to say “~1km high”

Reply: Done.

[57] suggest to replace “marked” with “interrupted” or “split-up”

Reply: Done.

[63] suggest to remove “-“ between “mountain” and “gaps”

Reply: Done.

[82] should be “model”

Reply: Done.

[103] should mention briefly here why focused on Tehuantepec and Papagayo but left the Panama gap as is

Reply: We omitted the Panama gap because it is a smaller gap than the Tehuantepec and Papagayo gap. Its impact on the TCs may also be minimal because no TCs (at least in the named storm category) have been generated in this region based on the available observations (see Figure below). We briefly mention this in the revised manuscript. Please see lines 139-140.

Figure R2-4: IBTrACS observed TCs. Colors indicate TC intensity.

[137-138] I don't understand what is meant by "Combining the opposite sign of changes together"

Reply: As can be seen in Figure 2 of revised manuscript, the TC genesis and track density changes are not homogeneous in the ENP. The changes in the open ocean area of the ENP are opposite to the changes in the nearshore area (eastern portion), but tend to dominate the net increase (decrease) in NMT (NGP).

[140] should be "mountains as well"

Reply: Done.

[142-143] Does the removal of these jets also explain the nmt decrease in TC activity in this region?

Reply: In NMT, since we removed the mountains, the entire regional circulation (not just the gap wind jets) was changed (please see the Figure R2-3g,m). We think the similar mechanism to NGP also applies to this case, that is, the dynamical role of moisture transport dominates the TC activity changes along the ENP coastline.

[154] You say “mountains can fuel TC genesis”. Is this backed up by the literature? Or would it be more accurate to say “mountain gaps can fuel TC genesis”?

Reply: Mountain gaps can fuel TC genesis through the injection of gap wind vorticities, and the lee vortex caused by the mountain itself can also contribute to the TC genesis (Zehnder 1991, Zehnder et al. 1999).

Zehnder, J. The interaction of planetary-scale tropical easterly waves with topography: A mechanism for the initiation of tropical cyclones. *J. Atmos. Sci.* **48**, 1217–1230 (1991).

Zehnder, J., Powell, D. M., & Ropp, D. L. The interaction of easterly waves, orography, and the intertropical convergence zone in the genesis of eastern Pacific tropical cyclones. *Mon. Wea. Rev.* **127**, 1566–1585 (1999).

[156] suggest to add “near” before “identical”

Reply: Done.

[157] suggest to replace “it provides” with “there is”

Reply: Done.

[165-166] suggest to move “well” before “represents”

Reply: Done.

[172] “terms” should be “term”

Reply: Done.

[179] suggest to move second apostrophe around the two pseudo phrases before the parentheses.

Reply: The change has been made.

[188] suggest to replace “which is directly resulting” with “which directly results”

Reply: Done.

[193] suggest to change “topographic precipitation” to “orographic precipitation”

Reply: Done.

[209] should say “resulting in more precipitation”

Reply: Done.

[229] suggest to remove “In particular”

Reply: Done.

[230] I think this would read clearer if you said “Removal of the mountains’ blocking effect”

Reply: Done.

[230] “diabetic” should be “diabatic” (here and anywhere else—I think I saw this one other place)

Reply: We apologize for this typo. Two changes have been made.

[232] suggest to replace “it” with “this teleconnection”

Reply: Done.

[295] “seasons” should be “season”

Reply: Done.

[304] What region is the nest performed over? Can you specify that here for clarity?

Reply: The nested domain coverage is now shown in Supplementary Figure 1a, which roughly covers the area [178°W-65°W, 2°S-36°N]. We have clarified this in the revised manuscript.

[311] suggest to remove “We specially note that”

Reply: Done.

[313] What is meant by “convexity” here? I’m not used to seeing this term describing topography.

Reply: Topographic convexity measures how convex or sharp the subgrid-scale topography is by statically relating the characteristics of the mountain waves to the subgrid-scale topography. It is an important parameter in the WRF topography-induced gravity wave drag parameterization.

[318] Can you briefly say how these relative vorticity thresholds are chosen?

Reply: $1.6 \times 10^{-4} \text{ s}^{-1}$ vorticity threshold is chosen for the 27-km simulation based on Knutson et al. (2007). Setting $1.6 \times 10^{-4} \text{ s}^{-1}$ as baseline, we carefully tuned the vorticity threshold for the 9-km simulation to optimally match the number of tropical cyclones in the CTL with observation (~17 in each JJASON season). After tuning, the averaged number of tropical cyclones in the 9-km simulation is 16.95 when the vorticity threshold is $4.8 \times 10^{-4} \text{ s}^{-1}$.

Knutson, T. R., Sirutis, J. J., Garner, S. T., Held, I. M. & Tuleya, R. E. Simulation of the recent multidecadal increase of Atlantic hurricane activity using an 18-km-grid regional model. *Bull. Am. Meteorol. Soc.* **88**, 1549–1565 (2007).

[322] “topography height” should be “topographic height”

Reply: Done.

[338] suggest to add “with observations” after “precipitation”

Reply: Done.

[338-352] Are all these biases generally consistent for both model resolutions? The comparisons shown in the SI are just for the 27 km model, right?

Reply: Yes, they are all similar but not identical. Note that 27 km simulations were performed for 29 individual seasons from 1990-2018, with 6-member ensemble for each season, but 9 km simulations were conducted in “climatology” mode. That is, we prescribed 1982-2018 climatology mean SST, while the lateral boundary conditions were derived from the 1996 season, which was characterized by a neutral phase of the Atlantic Multidecadal Oscillation. The initial conditions were from different dates from 1-30 April 1996 to generate 30-member ensemble. As such, the biases in the 9-km simulation cannot be bit-to-bit compared to those in the 27-km simulation since the two sets of experiments have different ensemble numbers and external forcing. As noted by Referee #3, the number of Supplementary Figures needs to be reduced for a better proportion, we only show the 9-km simulation biases in this reply as below.

Figure R2-5: Validation of the 9km CTL simulation. Seasonal averaged a) TC track density (occurrence number decade⁻¹), b) precipitation (mm day⁻¹), c) outgoing longwave radiation (OLR; W m⁻²), d) 10-m wind (m s⁻¹) and e) latent heat flux (W m⁻²) from the 9km CTL simulation 30-member ensemble mean. f-j) Are similar, but from various observations (refer subtitles for details).

[345-346] suggest to change to “This bias in South American orographic precipitation”

Reply: Done.

[348-349] suggest to “CCMPv2 and CTL shows significant consistency”

Reply: Done.

[359] should be “Papagayo”

Reply: Done.

[368] suggest to change to “a size of 5 zonal and 3 meridional gridcells and a standard deviation...”

Reply: This change has been made.

[396] “purposed” should be “proposed”

Reply: Done.

[400] suggest to remove “the” before “climate change”

Reply: Done.

[402] suggest to add “the” before “vertical and horizontal wind speeds”

Reply: Done.

[409] remove “that” before “determined by”

Reply: Done.

[416] shouldn't this be “determined from” not “defined by”? since you provide the actual definition on the next page

Reply: This change has been made.

[425] should have “is” before “determined”

Reply: Done.

[433] based on the prior line, shouldn't there be a factor of 2 in front of $(h_{10m}-h_{env})$?

Reply: Thanks for pointing this. We have changed to $b = (h_{10m}-h_{env}^*)/2$.

[440] “diabetic” should be “diabatic”

Reply: Done.

[448] suggest to write “; as such, we need”

Reply: Thanks for the suggestion. But based on Referee#3's comments we rewrote this subsection to reduce the length of Methods.

[449] add "such" before TCs

Reply: We rewrote this section in the revised manuscript.

[452] do you mean "that are adapted"?

Reply: Yes, but rewrote this section in the revised manuscript.

[456] add "the" before "95%"

Reply: We rewrote this section in the revised manuscript.

[492,498] "smoothened" should be "smoothed"

Reply: As of both your and Referee #3's comments, we have removed "CMIP6 Precipitation" in the revised manuscript.

[501] suggest to change to "similar to most subseasonal-to-seasonal prediction models;as such, the results"

Reply: We removed "CMIP6 Precipitation" in the revised manuscript.

[Fig 1 boxplots] What lat-lon region is the averaging performed over? Please specify in caption.

Reply: We average the seasonal TC activity in the area of 180°W to the North American coast, 0-30°N. We noted this in the Figure 2 caption of revised manuscript (Figure 1 of the initial submission).

[678] here and elsewhere you write "denoated". It should be "denoted"

Reply: Thanks for pointing our typo mistakes. We have corrected them in the revised manuscript.

[679] "experiment" should be "experiments"

Reply: Done.

[683,699] suggest to add "the" before "two-sided t-test"

Reply: We have added "the" before "two-sided t-test" in all places.

[Fig 2] Why do you use a different interval for the model vs ibtracs TC genesis density?Isn't this a bit misleading?

Reply: As the model simulation consists of a total 174 TC seasons while IBTrACS observation only has 29 TC seasons, the maximum value of simulated ensemble mean seasonal TC genesis density is consequently smaller than that in the observation, as ensemble mean tends to average out extreme values. Here we want to emphasize that the spatial pattern, rather than the amplitude, of TC genesis density shows agreement between CTL and IBTrACS observation, especially in the area between 120°W and 100°W. In the Figure 3 of revised manuscript, we labeled contour plots level for better clarity.

[719] should be “Sensitivity”

Reply: Thanks for pointing this.

[724] should be “quadrant”

Reply: Thanks for pointing this, we corrected it to “quadrant”.

[727] should be “activity”

Reply: Done.

[733] “minus” should be “subtracting”

Reply: Done.

[Fig 4a] Do the seasonal ensemble members for the CTL fall along this line too? If not, why not? I wasn't sure why that wasn't included in this figure.

Reply: As the seasonal TC activity in the ENP is not only controlled by the IVT following a linear relationship, the linear regression coefficient between the IVT and ENP TC days is statistically less significant than that shown in Figure 6a in the revised manuscript. Please see the attached plot below. Besides the Atlantic-Pacific interbasin moisture transport, ENP seasonal TC activity is also significantly modulated by tropical modes of climate variability, such as the Madden-Julian Oscillation and El Niño-Southern Oscillation, which can profoundly influence environmental vertical wind shear, deep convection, sea surface temperature, and mid-tropospheric moisture content. All these modulations can influence the environmental favorability for TCs (Maloney and Hartmann 2000; Chu 2004).

By subtracting the NMT/NGP with the corresponding season CTL 6-member ensemble mean to calculate the anomalies, as shown in Figure 6a, some of the influence from these modes of climate variability can be reduced, making the results cleaner for the influence of IVT. In addition, we concatenated the NMT and NGP anomalies to double the sample size when we determined the statistical confidence level.

Figure R2-6: Scatterplot of seasonal averaged cross basin integrated water vapor transport (IVT; $\text{kg m}^{-1} \text{s}^{-1}$) and seasonal accumulated number of days with the ENP TC activity. The seasonal averaged cross basin IVT is area averaged over the box of 95°W - 80°W , 10°N - 15°N cross the Central America. Binned scatterplots (large solid dots; intervals of $20 \text{ kg m}^{-1} \text{s}^{-1}$) with the linear regression line are overlaid.

Maloney, E.D. & Hartmann, D. L. Modulation of Eastern North Pacific hurricanes by the Madden–Julian oscillation. *J. Clim.* **13**, 1451–1460 (2000).

Chu, P-S. ENSO and tropical cyclone activity. *Hurricanes and Typhoons, Past, Present and Future*, R. J. Murnane and K.-B. Liu, Eds., Columbia University Press, 297–332 (2004).

[Fig 3] Is the black stuff at the bottom topography? Please explain what this is in the caption and how it is determined (ie meridional max, average, etc)

Reply: Yes, they are the averaged topographic height between 8°N and 15°N . We noted this in the Figure 5 caption of the revised manuscript.

[Fig S5] In the caption, you note panel c) but there is no panel c). Do you mean a or b?

Reply: Yes, thanks for pointing this.

[Fig S9] “neglectable” should be “negligible”

Reply: Done.

[Fig S10] is “changes” in initial bolded line a mistake?

Reply: We have added “TC activity” before “changes” for clarity. Now it is Supplementary Figure 8 in the revised manuscript.

[Fig S11] should be “distribution”

Reply: The change has been made.

[Fig S12] should be “Easterly waves”

Reply: Done. Note that we replace this figure with new Supplementary Figure 10 in the revised manuscript to better clarify the easterly waves changes in the ENP are primarily attributed to those easterly waves generated locally in the ENP, rather than transported from the Atlantic.

[Fig S13] How do you construct the high res and low res topography fields in panels g and h? Do you average the topography across models?

Reply: Yes, they are averaged from the available outputs from the CMIP6. Note that Supplementary Figure 13 was removed in the revised manuscript based on two Referees’ comments, as this part of analysis is less related to the current study.

[Fig S13] should say “purposes, precipitation on various”

Reply: Thanks for pointing this. Supplementary Figure 13 was removed in the revised manuscript.

[Fig S13] should say “Refer to Methods”

Reply: Thanks for pointing this. Supplementary Figure 13 was removed in the revised manuscript.

[Fig S14] What does TCM stand for? Also this is also a June-Nov average too, right? Should maybe specify this.

Reply: “TCM” stands for “tropical channel model”, which is the same model that we used in the CTL, NMT and NGP simulation. As our model is configured with a tropical channel domain, Fu et al. (2019) named it as TCM for short. We removed this figure in the revised manuscript.

Fu, D., Chang, P., Patricola, C. M. & Saravanan, R. High Resolution Tropical Channel Model Simulations of Tropical Cyclone Climatology and Intraseasonal-to-Interannual Variability. *J. Clim.* **32**, 7871-7895 (2019).

Point-to-point Reply to Referee #3

We would like to thank the referee for taking the time to review our manuscript and offering invaluable comments and suggestions to improve the manuscript. We have carefully followed each of your comments listed in red and revised our paper accordingly. Our replies to your comments are as follows:

Referee #3: This manuscript presents the interesting finding that tropical cyclone activity in the east north Pacific (ENP) is negatively affected by the topography of Central America, which is counter to some previous case studies. A novel aspect of this study is the use of a large climatological simulation period with a similarly large set of ensembles. Relatively course (27- km) and high resolution (9-km) simulations were carried out to show results were insensitive to resolution, though the higher resolution simulations are only included in supplementary material. A possible mechanism (moisture transport leading to increased buoyancy and ascent) is hypothesized for why removal of Central American terrain leads to a significant increase in TC activity in the ENP. There are a number of high quality figures that help illustrate the key results of this interesting manuscript.

Reply: Thank you for your encouraging comments.

Referee #3: However, I do have reservations about the structure of this manuscript. There are a large number of supplemental figures (14) which is more than three times than the total figures included in the main manuscript text (4). Moreover, these supplemental figures are referred to in both the results and methodology of this manuscript and seem more integral to the manuscript than just being included as supplemental material. This reviewer would like to see substantial revision of the manuscript, incorporating some of the more integral supplemental figures provided into the primary manuscript. There are also a few extraneous topics included in the methodology that seem only tangential to the main them of the manuscript (for example: how easterly waves and precipitation are affected by differences in

topography) that this reviewer would prefer to see incorporated into an additional manuscript.

Reply: Thanks for your comments. We have made significant changes in the structure of the revised manuscript. The paper now contains 6 figures in the main manuscript (compared to 4 in the previous version) and 10 supplementary figures (compared to 14 in the previous version). In the revision, the original Supplementary Figure 1 is now shown as Figure 1 in the main text, and the topic of how smoothed topography can influence precipitation in the CMIP6 is removed. As requested by Referee #1 and Referee #2, we kept the easterly wave related analysis in the revised manuscript since easterly wave is an important source of TC genesis, but rewrote the related methodology in the Method section, as well as reduced the length to make the revised manuscript more proportional in structure.

Referee #3: There is a lot of extraneous material in the methodology section that does not seem applicable to the manuscript theme which focuses on how Central American mountains can impact TC activity in the ENP. While the first four sections of the methods (High- resolution climate model simulations, Experiment designs, Modified genesis potential index, Buoyancy diagnostics) are all relevant to the manuscript, the final two sections (Easterly wave tracking and CMIP6 Precipitation; Lines 439-504) are too far removed from the manuscript theme to add overall value to this manuscript and seem like unnecessary extra material.

While it is true that easterly waves and precipitation are both meteorological phenomena that can influence ENP TC activity, the detailed description of the methods and results presented in the supplemental figures go beyond the scope of what this manuscript was inferred to discuss in the introduction (line 73-77).

The recommendation of this reviewer would be to remove this extraneous material that is already supplemental as organized by the authors. Specifically lines 439-504 and supplementary figures 12–14. These results and figures are high quality but would be more suitable for a subsequent manuscript.

Reply: Thanks for your thoughtful recommendation. Please refer to the reply above. We think the revised manuscript is now more balanced.

Referee #3: There are a number of instances where this reviewer would have preferred to structurally include the supplemental figures as regular figures in the text. For example, Supplemental figure 1 is a nice introductory figure that is the only figure discussed in the introduction. Why it is considered supplementary when it provides important background information provided in lines 52-67? There are a number of instances where it would have been preferable to include supplemental figures as regular figures in the manuscript and this will be occasionally described in the specific comments below.

Reply: We appreciate your thoughtful suggestion. As noted by Referee #1, this manuscript was initially submitted as a short letter that only allows a maximum of 4 figures. The original Supplementary Figure 1 was moved to the main text in the revised manuscript, and main text now has 6 figures and additional 10 figures are provided as supplementary.

L43: Instead of “tightly” use “closely” instead?

Reply: The change has been made.

L41-51: A few additional citations worth including in this portion of the introduction is the dramatic influence of convectively coupled Kelvin waves which have also been shown to have a significant influence on tropical cyclone development in the ENP.

Reply: Thank you for introducing these great works. We have added those references.

L82: “model” not “mode”

Reply: Done.

L105-107: This statement seems out of place with the rest of the paragraph. It might fit in better after the end of the sentence on line 85. Also can there be a quick explanation on why a tropical channel domain is better to handle mountain remote influences on TCs?

Reply: Thank you for this good suggestion. We have made the changes, please see lines 99-101.

Tropical channel domain enables the free circumnavigation of tropical waves and climate modes along the zonal direction. As shown in Supplementary Figure 8, the anomalous diabatic heating induced by the Central American mountain's blocking effect in the ENP can teleconnect with the other basins through equatorial wave propagation. Our results suggest that this teleconnection only results in modest and insignificant changes in TC activity over the other TC-active regions. Conventional regional models require external forcing at western and eastern boundaries, which would most likely be derived from the existing reanalysis datasets. The simulation results could be biased if the model was constrained by zonal boundary conditions, as reanalysis datasets do not include the effect of the modified Central American mountains on atmosphere circulation changes.

L108-117: This reviewer very much appreciates that the 27-km simulation results are being compared to the 9-km simulations described in this portion of the results. But I would have preferred to see more details comparing the two simulations in the following section (lines 146- 153) especially given the knowledge higher resolution would be more likely to capture important mesoscale circulations produced by the terrain gaps and have better resolution to capture TC circulation and wind field aspects not touched on in this manuscript.

Reply: We agree that 9-km simulation is more capable to resolve TC internal dynamics than 27-km and it is very intriguing to disentangle how Central American mountains can influence the ENP TC structures. As noted by Supplementary Figure 9 (also shown below), the probability density functions (PDFs) of TC maximum 10-m

wind and minimum sea level pressure do show some differences, while the differences are not homogeneous. For example, in weak-to-moderate TC intensity regime (i.e. > 23 m/s but < 40 m/s or < 1005 hPa but > 970 hPa), the fraction in NMT is apparently more than that in CTL and NGP, but this difference reverses when we focus on strong TC intensity regime (i.e. > 40 m/s or < 970 hPa). However, as these differences in TC intensity is statistically less significant than the seasonal mean TC activity metrics (i.e. TC numbers, TC days) differences, and the main scope of this manuscript is to reveal the opposite role of Central American mountains played in hindering the seasonal ENP TC activity. We do not have enough space to fit the discussion of the TC structural changes to this short paper in Nature Communications. We plan to make more comprehensive studies about the TC dynamics, including the composite analysis of all simulated TCs (more than 510, 690 and 390 individual TC cases in the 9-km CTL, NMT and NGP, respectively) from both axisymmetric and asymmetric point of views, for future publications in more technical-oriented journals, such as *Journal of Atmospheric Sciences*.

Figure R3-1: a) Maximum 10-m wind velocity (m s^{-1}) of TCs from the CTL (black solid line), NMT (red solid line), NGP (blue solid line), CTL9km (black dashed line), NMT9km (red dashed line), and NGP9km (blue dashed line). The peak intensity within each experiment is shown in the legend. b) Similar, but for TC intensity in terms of minimum sea level pressure (hPa).

L141-142: Use “Gulf of Tehuantepec” instead of just Tehuantepec?

Reply: The change has been made.

L142-143: Also reference supplemental figure 1 here since you are referring to the jet curvature seen in Figs. 1 and 4? This is another instance where the supplemental figures would be better off being in the primary manuscript.

Reply: Supplementary Figure 1 is now moved as the Figure 1 in the main manuscript.

L172-178: This passage nicely explains how GPI is broken down into its individual components but is another example where the supplemental figure 7 could have been

moved to right after Fig. 2 in the primary manuscript. Instead the reader is forced to hunt town this important figure for this passage in another location. Structurally this makes the manuscript more difficult to read.

Reply: Thanks for pointing this. We moved original Supplementary Figure 7 (GPI contributions) to main text as Figure 4 in the revised manuscript. We hope this can make the readers easier to follow.

L180 and in other places: “state-of-art” is superfluous.

Reply: We have deleted all three of them in the revised manuscript.

L204-214: Missing in this TC genesis link here is how enhanced buoyancy and moisture results in vorticity spin up necessary for tropical cyclogenesis. I agree with the authors that higher vertical velocity and buoyancy via enhanced moisture is likely to result in more favorable conditions for tropical cyclogenesis but need to also describe how stronger deep moist ascent also results in vortex stretching and aggregation of pre-existing larger-scale vorticity. See citations below:

Davis, C., C. Snyder, and A. C. Didlake, 2008: A Vortex-Based Perspective of Eastern Pacific Tropical Cyclone Formation. *Mon. Weather Rev.*, **136**, 2461–2477, <https://doi.org/10.1175/2007MWR2317.1>.

Wang, Z., 2014: Role of Cumulus Congestus in Tropical Cyclone Formation in a High- Resolution Numerical Model Simulation. *J. Atmos. Sci.*, **71**, 1681–1700, <https://doi.org/10.1175/JAS-D-13-0257.1>.

Reply: Thanks very much for introducing these references. We have added them in the revised manuscript, please see lines 256-258. Meanwhile, we added the below figure as Supplementary Figure 4 to better illustrate how the Central American mountains induced mean circulation changes in the ENP can result in positive vorticity anomalies that can influence seasonal ENP TC activity.

Figure R3-2: Topography-induced responses of mean circulation. Seasonal averaged a) 925hPa wind speed (m s^{-1}), b) 925hPa horizontal wind divergence (10^{-5} s^{-1}), c) 925hPa relative vorticity (10^{-5} s^{-1}), d) 925 hPa vorticity equation vortex stretching term $[-(f + \zeta)(\partial u/\partial y + \partial v/\partial x); 10^{-10} \text{ s}^{-2}]$, e) column integrated water vapor (IWV; mm), and f) 500 hPa vertical wind (cm s^{-1}) from CTL simulation ensemble mean. g-l) And m-r) are similar, but from the differences using the NMT minus CTL and NGP minus CTL, respectively. Grey dots denote statistical confidence at the 95% level based on the two-sided t-test. Yellow sketchings indicate the mountain outlines at 925 hPa.

L230: Diabatic... not diabetic

Reply: We apologize for this reckless typo. The changes have been made.

L336: Is this using the CTL simulation in the 27-km domain or the 9-km domain? Also this information seems less about the methods, but more about verifying the control simulation with reanalysis and observation datasets. This would have made for a nice section in the results before getting into the main simulation comparisons using different terrain types. This comment could also be applied to the discussion related to Supplemental Figures 4-6.

Reply: The comparisons shown in original Supplementary Figures 3-6 are between 27-km simulation and observation. Please refer to the below figure for 9-km results, and Supplementary Figure 1 in the revised manuscript for 27-km results. We have moved L334-L353 to Section “High-resolution tropical cyclone permitting climate simulation” in the revised manuscript, please see lines 113-130.

Figure R3-3: Validation of the 9km CTL simulation. Seasonal averaged a) TC track density (occurrence number decade⁻¹), b) precipitation (mm day⁻¹), c) outgoing longwave radiation (OLR; W m⁻²), d) 10-m wind (m s⁻¹) and e) latent heat flux (W m⁻²) from the 9km CTL simulation 30-member ensemble mean. f-j) Are similar, but from various observations (refer subtitles for details).

L397: Again state-of-art is superfluous and not necessary here.

Reply: The change has been made.

L439-504: This portion of the methods section seems unnecessarily to the rest of the manuscript. How the CTL simulation depicts easterly waves and precipitation patterns is likely an important discussion but seems beyond the scope of this study that already includes 18 figures (14 supplementary). Recommend cutting this portion of the manuscript as discussed in major comment #1.

Reply: As noted above, we much appreciate your suggestions on the shape of manuscript. The topic of how smoothed topography could influence precipitation in the CMIP6 is removed in the manuscript. But as arises from the comments by Referee #1 and #2, we still remain the easterly wave related analysis. We have reduced the total number of figures to 16 (6 in the main text and 10 in the supplementary) in the revised manuscript.

Reviewer comments, second round –

Reviewer #2 (Remarks to the Author):

Overall evaluation: Recommend accept pending minor revisions

The authors have done an excellent job addressing my prior questions/ concerns. I especially appreciate the added moisture budget analysis, and their efforts to focus the narrative of the paper. I recommend the paper be accepted, but highlight below some suggestions to make the moisture budget analysis clearer (especially the comment directly below), and some various typos/grammatical errors.

[Eq 6, SI Fig. 6] I have trouble corresponding these equation components to the labels in SI Fig. 6c,f x-axes. Perhaps you could just label the x-axis directly with the equation components instead? Or otherwise group into simply transient eddy, dynamical (monthly), thermal (monthly), and cross term (monthly) terms as in Baldwin and Vecchi 2016 Fig. 8b.

Minor text revisions:

1. [68] suggest to change to "generate" rather than "exert"
2. [77-78] I think you mean "these systematic model biases can also be partially attributed to the poor topographic representation"
3. [80] should be "one natural question arises"
4. [81] suggest to change to "lead to biases in the ENP"
5. [96] should be "an atmosphere-ocean coupled model"
6. [117] "overestimate" what? I think you're missing something here.
7. [121] suggest to change to "weakened precipitation patch"
8. [122-123] Suggest to change to "The near-surface circulation pattern over the Caribbean Sea and the ENP is consistent between satellite observations and CTL"
9. [129] suggest to soften to "helps ensure"
10. [139] suggest to change to "We do not make changes at the Panama Gap"
11. [253] suggest to change to "extension of moisture transport by the seasonal mean winds from the Caribbean Sea..."
12. [255] suggest to add "of" before "the moisture content"
13. [257] typo should be "stretching"
14. [257-8] suggest to change to "environmental conditions more (less) favorable for TC genesis"
15. [291] suggest to remove "our" since it's the whole field's fundamental understanding you're advancing
16. [342] should be "summer seasons"
17. [392] punctuation issue should be "; as such,"
18. [409] should be "rectangular area of surface elevations was modified"
19. [432] should be "state-of-the-art"
20. [467] should have "is" after "h10m-h*env"
21. [471] suggest to change to "water vapor is unsaturated"
22. [477] I think you meant to cite 53-56 here.
23. [492] should be "involve" since "contributions is plural"
24. [493] should be "thermodynamic contributions, which only involve" to be consistent with "dynamic contributions"
25. [504] Citation style is different here than in the rest of the paper.
26. [528] should be "the Atlantic"
27. [531] For the authors' reference (eg for future publications no worries for this one), I recently had to upload a substantial amount of hi-res model data to a public repository (I used Zenodo) for another journal, and it might be worth looking into as a way to make some subset of these model simulation diagnostics available to the broader community and citable. A challenge is of course the high temporal resolution of the data in this paper, but I know Zenodo can grant permission for datasets greater than the standard 50GB limit if you email them. This command line method to upload to Zenodo was a life-saver for me in doing this if you ever go this route:
<https://gist.github.com/maxogden/b758cf0fe6d353846ef9ce7d03fdca0c>

28. [SI Fig 4] suggest to describe as "light orange shading" rather than "yellow sketching"
29. [SI Fig 5] typo should be "moisture"

Reviewer #4 (Remarks to the Author):

Review of "Central American mountains inhibit eastern North Pacific seasonal tropical cyclone activity" by Fu et al.

Recommendation: Minor revisions

Summary:

This is my first review of this manuscript, and I was also asked to step in to assess whether the comments of Reviewer 1 were addressed. This study argues that the mountains of Central America hinder east Pacific tropical cyclogenesis by suppressing the moisture flux into the east Pacific from the Atlantic. This result suggests that models need to properly represent the mountains of Central America in order to represent and predict cyclogenesis in the east Pacific.

This is an interesting and relevant study that deserved publication Nature Communications. I feel that the authors have done an adequate job in responding to previous reviewer comments, especially to Reviewer 1, and I just have some additional minor comments for the authors to consider in revising the paper. I recommend that this paper be published after minor revisions.

Minor Comments:

- 1) line 102. Stating that three sets of simulations will be conducted, but providing no information on the nature of these simulations until later, is a bit awkward. Possibly a brief introduction can be given here to the differences between the simulations.
- 2) lines 113-117. A more faithful discussion of the errors in control precipitation relative to the observed dataset should be conducted. In particular, the control simulation produces too much precipitation to the west of the mountains, and the precipitation minimum in the Costa Rica Dome region is generally non-existent or in the wrong place. These results can significantly affect the conclusions about cyclogenesis.
- 3) lines 163-164. Whitaker and Maloney (2020) suggest that the Papagayo jet might help to spin up the synoptic precursor disturbances for tropical cyclones. Could this signal near the coast in NMT reflect the loss of that jet feature when getting rid of the mountains?
- 4) line 204. It is not clear why "NGP" is in parentheses here.
- 5) lines 204-208. One limitation of the experiments here is that even though some experiments remove the impact of wind jets, the SST minimum signature in the Costa Rica Dome region in these experiments is not removed and still exists, hence creating inconsistency between the SST boundary condition used and the wind patterns. Some discussion of this inconsistency and implications for the TC results might be useful.
- 6) Figure 5. The difference in longitudes between the top row and the middle and bottom rows makes the comparison between the two difficult. The authors might consider using the same x-axis.
- 7) line 238-239. Again, this is a situation where the Costa Rica Dome SST minimum might have an impact. If this SST minimum went away with removal of the gap winds, how might this analysis of buoyancy change?
- 8) lines 255-256. What is meant by "moisture convection"?
- 9) line 270-271. I would hesitate to make such a definitive statement on causality here, although

the analysis is suggestive. There are other factors that are affected by removing topography that may also show a relationship to TC days that are as strong as cross-basin IVT transport.

10) line 284-285. This is another place where reference to Whitaker and Maloney (2020) would be useful. Rydbeck and Maloney (2017) also showed an influence of removing topography in Central and South America on easterly wave generation via suppressing the diurnal cycle that could be worth a mention.

Revision Note NCOMMS-20-31758A

“Central American mountains inhibit eastern North Pacific seasonal tropical cyclone activity” by Dan Fu, Ping Chang, Christina M. Patricola, R. Saravanan, Xue Liu, and Hylke E. Beck.

Point-to-Point Reply to Referee #2

We would like to thank the referee again for the invaluable comments and help us improve the manuscript. We have carefully followed each of your comments listed in red and revised our paper accordingly. Our replies to your comments are as follows:

Referee #2:The authors have done an excellent job addressing my prior questions/concerns. I especially appreciate the added moisture budget analysis, and their efforts to focus the narrative of the paper. I recommend the paper be accepted, but highlight below some suggestions to make the moisture budget analysis clearer (especially the comment directly below), and some various typos/grammatical errors.

Reply: We appreciate these positive comments.

[Eq 6, SI Fig. 6] I have trouble corresponding these equation components to the labels in SI Fig. 6c,f x-axes. Perhaps you could just label the x-axis directly with the equation components instead? Or otherwise group into simply transient eddy, dynamical (monthly), thermal (monthly), and cross term (monthly) terms as in Baldwin and Vecchi 2016 Fig. 8b.

Reply: We have taken your suggestion and inserted the equation components to the bar plot for better clarity purpose. Please see the revised Figure S6 (also attached below).

Supplementary Figure 6 | Moisture budget decomposition. The differences of moisture flux convergence a) by the mean flow ($-\frac{1}{g\rho_w} \int_0^{P_S} \nabla \cdot (\bar{\mathbf{u}}\bar{q}) dp$; unit: mm day⁻¹; colorshadings) and b) by the transient eddy ($-\frac{1}{g\rho_w} \int_0^{P_S} \nabla \cdot \overline{(\mathbf{u}'q')}$; unit: mm day⁻¹; colorshadings) using the NMT minus CTL ensemble mean. The precipitation minus evaporation difference (P-E; mm day⁻¹) is overlaid in black contours (contour starting from -2.5 with the interval of 1; positive solid lines and negative dotted lines). c) Relative contributions of various moisture budget terms to changes in hydrological cycle averaged over the red box (120°W to 93°W, 8°N to 15°N) in panel a). Refer to Methods for details of moisture budget analysis and decomposition of moisture flux convergence. d-f) Similar to a-c), but for the NGP minus CTL. Grey dots denote statistical confidence at the 95% level based on the two-sided t-test.

Minor text revisions:

1. [68] suggest to change to “generate” rather than “exert”

Reply: Done.

2. [77-78] I think you mean “these systematic model biases can also be partially

attributed to the poor topographic representation”

Reply: Thanks. Changes are made.

3. [80] should be “one natural question arises”

Reply: Done.

4. [81] suggest to change to “lead to biases in the ENP”

Reply: Thanks. The change has been made.

5. [96] should be “an atmosphere-ocean coupled model”

Reply: Done.

6. [117] “overestimate” what? I think you’re missing something here.

Reply: Many climate models overestimate ITCZ strength, and even show biased location towards northward.

7. [121] suggest to change to “weakened precipitation patch”

Reply: Thanks. The change has been made.

8. [122-123] Suggest to change to “The near-surface circulation pattern over the Caribbean Sea and the ENP is consistent between satellite observations and CTL”

Reply: Thanks. The change has been made.

9. [129] suggest to soften to “helps ensure”

Reply: We thank Referee for this suggestion. We have it changed to “helps ensure”.

10. [139] suggest to change to “We do not make changes at the Panama Gap”

Reply: Done.

11. [253] suggest to change to “extension of moisture transport by the seasonal mean winds from the Caribbean Sea...”

Reply: Done.

12. [255] suggest to add “of” before “the moisture content”

Reply: Done.

13. [257] typo should be “stretching”

Reply: Thanks. The change has been made.

14. [257-8] suggest to change to “environmental conditions more (less) favorable for TC genesis”

Reply: The change has been made.

15. [291] suggest to remove “our” since it’s the whole field’s fundamental understanding you’re advancing

Reply: Done.

16. [342] should be “summer seasons”

Reply: Done.

17. [392] punctuation issue should be “; as such,”

Reply: Done.

18. [409] should be “rectangular area of surface elevations was modified”

Reply: Done.

19. [432] should be “state-of-the-art”

Reply: The change has been made.

20. [467] should have “is” after “h10m-h*env”

Reply: Done.

21. [471] suggest to change to “water vapor is unsaturated”

Reply: Done.

22. [477] I think you meant to cite 53-56 here.

Reply: Done.

23. [492] should be “involve” since “contributions is plural”

Reply: Done.

24. [493] should be “thermodynamic contributions, which only involve” to be consistent with “dynamic contributions”

Reply: Done.

25. [504] Citation style is different here than in the rest of the paper.

Reply: Done.

26. [528] should be “the Atlantic”

Reply: Done.

27. [531] For the authors’ reference (eg for future publications no worries for this one), I recently had to upload a substantial amount of hi-res model data to a public repository (I used Zenodo) for another journal, and it might be worth looking into as a way to make some subset of these model simulation diagnostics available to the broader

community and citable. A challenge is of course the high temporal resolution of the data in this paper, but I know Zenodo can grant permission for datasets greater than the standard 50GB limit if you email them. This command line method to upload to Zenodo was a life-saver for me in doing this if you ever go this route: <https://gist.github.com/maxogden/b758cf0fe6d353846ef9ce7d03fdca0c>

Reply: We thank the Referee for this useful information.

28. [SI Fig 4] suggest to describe as “light orange shading” rather than “yellow sketching”

Reply: We have changed it to “Light orange shadings”

29. [SI Fig 5] typo should be “moisture”

Reply: Thanks. The change has been made.

Point-to-Point Reply to Referee #4

We would like to thank the referee for taking the time to review our manuscript and offering invaluable comments and suggestions to improve the manuscript. We have carefully followed each of your comments listed in red and revised our paper accordingly. Our replies to your comments are as follows:

This is my first review of this manuscript, and I was also asked to step in to assess whether the comments of Reviewer 1 were addressed. This study argues that the mountains of Central America hinder east Pacific tropical cyclogenesis by suppressing the moisture flux into the east Pacific from the Atlantic. This result suggests that models need to properly represent the mountains of Central America in order to represent and predict cyclogenesis in the east Pacific.

This is an interesting and relevant study that deserved publication Nature Communications. I feel that the authors have done an adequate job in responding to previous reviewer comments, especially to Reviewer 1, and I just have some additional minor comments for the authors to consider in revising the paper. I recommend that this paper be published after minor revisions.

Reply: Thank you for your encouraging comments.

Minor Comments:

1) line 102. Stating that three sets of simulations will be conducted, but providing no information on the nature of these simulations until later, is a bit awkward. Possibly a brief introduction can be given here to the differences between the simulations.

Reply: We thank you for these constructive comments. We have followed your suggestion and added a brief introduction about these simulations, please see lines 102-105.

2) lines 113-117. A more faithful discussion of the errors in control precipitation relative to the observed dataset should be conducted. In particular, the control simulation produces too much precipitation to the west of the mountains, and the

precipitation minimum in the Costa Rica Dome region is generally non-existent or in the wrong place. These results can significantly affect the conclusions about cyclogenesis.

Reply: We thank the Referee for this suggestion. The figure below compares 1990-2014 JJASON mean precipitation from the observations with various climate model simulations (including the CTL described in this study). It is clear that, many climate models suffer from unrealistic ITCZ simulations in the ENP even in the most recent CMIP6 project. None of these state-of-the-art high-resolution climate models can faithfully reproduce the observed precipitation minimum in the Costa Rica Dome region and most of these models have positive precipitation biases on the leeside of the mountain. Relatively speaking, CTL appears to be one of the best (just behind MRI-AGCM3-2-S), albeit still imperfect, among all these latest models in terms of simulating overall precipitation in this region. We attribute this to the careful turning of model parameterizations, as described in details in Fu et al. (2019).

Figure R4-1: Comparison of MSWEPv2.2 June-November 1990-2014 averaged precipitation (unit: mm/day) with 27km CTL (top middle column) and various CMIP6 HighResMIP atmosphere-only GCM simulation with the highresSST-present forcing (observed SST during 1950-2014). Horizontal resolution of each model is listed for reference.

To investigate uncertainty in the responses of TCs to the Central American mountains due to model horizontal resolution and finer-scale topography, we performed higher-resolution downscaled CTL, NMT and NGP simulations, in which a 9 km subdomain is one-way nested within the 27 km tropical channel over the ENP and Gulf of Mexico. We note that, the precipitation minimum in the Costa Rica Dome region is better captured in the 9 km nested CTL simulation, and the positive precipitation bias west of the Central American mountains in 27km CTL is largely reduced as well (Supplementary Figure 3a), as shown in the figure below. We emphasize that these 9 km nested simulations produce nearly identical results in terms of percentage changes of TC activity in response to the topography modifications, with TC activity increased by about 36% (35% in 27km) in 9km NMT and decreased by 25% (22% in 27km) in 9km NGP simulations. Please refer lines 142-150 and lines 178-184 and Supplementary Figure 3 for more details. Thus, based on the results of these higher resolution simulations that reduce the precipitation biases in the vicinity of Costa Rica Dome but produce nearly identical TC responses, we believe the conclusion that Central America mountains inhibit ENP seasonal TC activity is robust.

Figure R4-2: Comparison of MSWEPv2.2 June-November 1990-2018 averaged precipitation (unit: mm/day) with 30-member ensemble mean 9km CTL.

Fu, D., Chang, P., Patricola, C. M. & Saravanan, R. High Resolution Tropical Channel Model Simulations of Tropical Cyclone Climatology and Intraseasonal-to-Interannual Variability. *J. Clim.* **32**, 7871-7895 (2019).

3) lines 163-164. Whitaker and Maloney (2020) suggest that the Papagayo jet might help to spin up the synoptic precursor disturbances for tropical cyclones. Could this signal near the coast in NMT reflect the loss of that jet feature when getting rid of the mountains?

Reply: From Supplementary Figure 4g, we can see that although the Papagayo jet is eliminated in the NMT simulation, the Tehuantepec jet is significantly strengthened. As a result, the net change in low-level relative vorticity near the Papagayo jet region (86°W-89°W, 9°N-13°N based on Whitaker and Maloney 2020) cannot be simply attributed to the change in the Papagayo jet (Supplementary Figure 4i). By comparing the simulated easterly waves between NMT and CTL (Supplementary Figure 10b and 10e), we find that the easterly waves are suppressed near the Papagayo jet region. Thus, we think our results are qualitatively consistent with the findings of Whitaker and Maloney (2020).

4) line 204. It is not clear why “NGP” is in parentheses here.

Reply: By assessing the changes in Genesis Potential Index, we find that the environmental favorability for TCs is increased in the NMT simulation, but decreased in the NGP simulation. To save space, we want to state these opposite-sign changes in one sentence, and thus we parenthesize “less” and “NGP” in the sentence.

5) lines 204-208. One limitation of the experiments here is that even though some experiments remove the impact of wind jets, the SST minimum signature in the Costa Rica Dome region in these experiments is not removed and still exists, hence creating inconsistency between the SST boundary condition used and the wind patterns. Some discussion of this inconsistency and implications for the TC results might be useful.

Reply: Thank you for your comment. First of all, we want to correct an embarrassing mistake in original Figure 1a – **we erroneously plotted the January SST climatology**, instead of the June-November seasonal mean. In the revision, we are now showing in white contours the June-November mean SST averaged over 1990-2018.

We agree that the inconsistency between the prescribed SSTs and the simulated wind patterns in the ENP near-coastal area is a limitation of this study. However, we hypothesize its influence on seasonal TC activity may not be significant, at least in terms of seasonal accumulated number of TCs and TC days, due to the following reasons:

- 1) After using the correct SST climatology, we find that the cold patch near the Costa Rica dome is quite weak in the boreal summer (see Figures below). Along 10°N, the SST difference is less than 0.5°C at the Costa Rica Dome (around 88°W, 10°N), and the SST remains above 27.5°C. This is consistent with Figure 8 of Xie et al. (2005). During the boreal summer, the Costa Rica Dome is characterized by a shallower ocean thermocline rather than a strong cold SST patch (Figure 4 of Xie et al. 2005).
- 2) During the boreal summer, the Costa Rica thermocline Dome is the result of ocean upwelling associated with wind stress curl from the Papagayo jet and

additionally the positive curl of the northward extended ITCZ (Xie et al. 2005). In the NMT (and also NGP) simulation, although the Papagayo jet is eliminated (Supplementary Figure 4g and 4m), the mean position of the ITCZ is also changed concurrently. Combined, the net change in low-level relative vorticity is complex – both NMT and NGP show positive relative vorticity anomalies north of 10°N but negative vorticity anomalies lie south (Supplementary Figure 4i and 4o). Thus, the Costa Rica thermocline Dome may not be removed in a coupled NMT and NGP experiments, but its location can be displaced westward or northward.

- 3) In addition, both the observations and CTL simulation suggest that ENP TC activity is relative weak east of 90°W (Figure 3a,d and Figure 2f, i). As the simulated TC responses to the modifications in the Central American mountains are predominately confined to June-September (Figure 6b and 6c) and over a broader offshore area (west of 100°W; Figure 2g,h and Figure 2j,k) of ENP, the influence from the weak SST cold patch (less than 0.5°C) in the Costa Rica Dome (around 88°W, 10°N) on the seasonal TC activity over open-ocean may be insignificant. Note that, the TC activity anomalies east of 95°W are negative (insignificant) in the NMT (NGP) simulation (Figure 2g,h,j,k and Supplementary Figure 2h,i,k,l), which are opposite to the TC activity anomalies over the open-ocean and contribute negatively to the net TC responses in the ENP.

The above factors lead us speculate that the lack of atmosphere-ocean coupling over the Costa Rica Dome region in our study may not fundamentally change our conclusion. However, we completely agree that future studies are needed to investigate the effects of atmosphere-ocean interactions in these simulations. As TCs can induce ocean surface cooling and subsurface warming through changing upper ocean vertical mixing, the changes in TC activity in response to topography modifications can potentially lead to changes in the upper ocean thermocline structure, which in turn may feedback to the regional Hadley Cell and TC environmental favorability. Future coupled climate model studies can help to quantify this issue. Some discussions of the lack of atmosphere-ocean coupling in this study are made in lines 311-316.

Xie, S., Xu, H., Kessler, W. & Nonaka, M. Air–Sea Interaction over the Eastern Pacific Warm Pool: Gap Winds, Thermocline Dome, and Atmospheric Convection*. *J. Clim.* **18**, 5-20 (2005).

Figure R4-3: January-March (left) and June-November (right) 1990-2018 averaged NOAA Optimum Interpolation SST (unit: °C). SSTs along 10°N are showed in the bottom row.

6) Figure 5. The difference in longitudes between the top row and the middle and bottom rows makes the comparison between the two difficult. The authors might consider using the same x-axis.

Reply: We have re-plot Figure 5 using the same x-axis.

7) line 238-239. Again, this is a situation where the Costa Rica Dome SST minimum might have an impact. If this SST minimum went away with removal of the gap winds, how might this analysis of buoyancy change?

Reply: Please see our reply to 5). Based on the June-November seasonal mean SST (Fig. R4-3), we believe the cold SST in the Costa Rica Dome during June-November is too weak to have a significant influence on TC activities. However, a full understanding of this issue will require future coupled climate model simulations as we discussed in the revision (please see lines 311-316).

8) lines 255-256. What is meant by “moisture convection”?

Reply: We apologize for the typo; it should be “moist convection”. By the AMS Glossary of Meteorology definition, moist convection is the type of atmospheric convection in which the phase changes of water play an appreciable role.

9) line 270-271. I would hesitate to make such a definitive statement on causality here, although the analysis is suggestive. There are other factors that are affected by removing topography that may also show a relationship to TC days that are as strong as cross-basin IVT transport.

Reply: We apologize for the misleading. We want to state, “The cross-basin IVT changes are unlikely caused by the TC activity changes”. This is because we carefully selected the upstream IVT averaging region (95°W-80°W and 10°N-15°N) where TC activity changes are insignificant. Therefore, we reason it is the upstream cross-basin IVT changes cause downstream TC activity changes. We totally agree that cross-basin IVT is not the only factor to control the ENP TC activity, and we rephrased “it is evident that ...” to “it indicates that ...”.

10) line 284-285. This is another place where reference to Whitaker and Maloney (2020) would be useful. Rydbeck and Maloney (2017) also showed an influence of removing topography in Central and South America on easterly wave generation via suppressing the diurnal cycle that could be worth a mention.

Reply: Thank you for introducing these relevant works. We have added Whitaker and Maloney (2020) to the revised manuscript.

Whitaker, J. W., & Maloney, E. D. Genesis of an East Pacific easterly wave from a Panama Bight MCS: A case study analysis from June 2012. *J. Atmos. Sci.*, **77**, 3567–3584 (2020).

Reviewer comments, third round –

Reviewer #4 (Remarks to the Author):

The authors have done a thorough job of addressing my previous comments, and I thus recommend acceptance of this manuscript.